



# Seasonal variability in mass, nutrients and DOC lateral transports off Northwest African Upwelling System

Nadia Burgoa[1], Francisco Machín[1], Ángeles Marrero-Díaz[1], Ángel Rodríguez-Santana[1],
Antonio Martínez-Marrero[2], Javier Arístegui[2], and Carlos M. Duarte[3]

[1]Departamento de Física, Universidad de Las Palmas de Gran Canaria, Spain
[2]Instituto de Oceanografía y Cambio Global, Universidad de Las Palmas de Gran Canaria, Spain
[3]Red Sea Research Center, King Abdullah University of Science and Technology, Saudi Arabia

**Correspondence:** Nadia Burgoa (nadia.burgoa@ulpgc.es)

**Abstract.**

The Coastal-Ocean Carbon Exchange in the Canary Region Project (COCA) arises in order to analyse and get to understand the impact of lateral export of nutrients and organic matter from the highly productive Coastal Upwelling System off NW Africa in the biogeochemical cycles during two different seasons.

5     The circulation patterns off NW African Upwelling System are examined by applying an inverse model to two hydrographic datasets gathered in fall 2002 and spring 2003. The mass transports estimated by model are consistent with the thermal wind equation and the conservation of mass in a closed volume. Besides, the Ekman transport and the freshwater flux are also considered.

    These estimates show a seasonal variability in the circulation patterns at central levels, particularly in the southern boundary 10 of the domain, where the Cape Verde Frontal Zone is located. In the beginning of fall, this circulation is deeper and northward with a net transport of $6 \pm 3$ Sv and, in the late spring, it is shallower and southward with a similar intensity. At intermediate levels important differences are also observed between the two seasons. In fall, the Antarctic Intermediate Waters reaches higher latitudes with $2 \pm 2$ Sv flowing northward. During spring, there is no significant northward flow of AAIW. However, there is a moderate westward mass transport which impacts both the lateral transports of inorganic nutrients and organic matter 15 at intermediate layers and also the shallowest lateral transports of organic matter.

    Seasonal variability in circulation patterns are also reflected in lateral transports of inorganic nutrients and dissolved organic carbon. Therefore, the changes in the circulation patterns between the two seasons have allowed us to assess the variability in the contributions of $SiO_2$, $NO_3$, $PO_4$ and DOC from the first to the second season. In fall, the transports are mainly northward from the south with $-0.80 \pm 0.34$, $-1.11 \pm 0.47$ and $-0.07 \pm 0.03 \, \mathrm{kmol\,s^{-1}}$ of $SiO_2$, $NO_3$ and $PO_4$, respectively. 20 In spring, however, lateral transports off-shore are favoured with $0.75 \pm 0.37$, $1.34 \pm 0.66$ and $0.08 \pm 0.04 \, \mathrm{kmol\,s^{-1}}$ of $SiO_2$, $NO_3$ and $PO_4$, respectively. This westward transport stimulates in turn an intensified westward DOC transport at shallow layers, specifically $0.50 \pm 0.25 \times 10^8 \, \mathrm{mol\,C\,day^{-1}}$.



# 1  INTRODUCTION

The North Atlantic Subtropical Gyre (NASG) is one of the most important components in the thermohaline circulation.
It presents a well-known intensification in its western margin, the Gulf Stream, with maximum velocities up to $2\ \mathrm{m\,s^{-1}}$
(Halkin et al., 1985). The currents observed in this western margin of the gyre occupy a small horizontal extension as compared to that of the currents in the eastern side, resulting in an asymmetric gyre (Stramma, 1984; Tomczak and Godfrey, 2003).
The low intensity of the currents at the eastern boundary made them very little studied until the 1970s, when CINECA program focused on the productive African upwelling (Ekman, 1923; Tomczak, 1979; Hughes and Barton, 1974; Hempel, 1982).
Käse and Siedler (1982) found striking intense currents south of the Azores connected to the Gulf Stream and suggested that part of the recirculation of the NASG occurs southward in the vicinity of the African coast. Later on, several surveys based on both *in situ* and remote sensing observations contributed to define the general characteristics for the average flow of the region
(Käse and Siedler, 1982; Stramma, 1984; Käse et al., 1986; Stramma and Siedler, 1988; Mittelstaedt, 1991; Zenk et al., 1991;
Fiekas et al., 1992; Hernández-Guerra et al., 1993).

Most of the eastward flow from the Gulf Stream is confined to a band between the Azores and Madeira Islands, recirculating southward through the Canary Islands and north of the Cape Verde Islands to become into a southwestward flow (Stramma,
1984). This current system is composed by the Azores Current (AC), the Canary Current (CC), the Canary Upwelling Current
(CUC), the North Equatorial Current (NEC) and the Poleward Undercurrent (PUC). The AC divides into several branches defining the boundary current system off Northwest Africa. It firstly feeds the Iberian Current (Haynes et al., 1993) while a second significant branch enters the Mediterranean Sea (Candela, 2001). Most of the AC recirculates southward splitting into the main CC across the Canarian archipelago and the secondary CUC (Pelegrí et al., 2005, 2006). These currents extend southward developing the Cape Verde Frontal Zone (CVFZ), a density-compensated front with North Atlantic Central Water at its northern side and South Atlantic Central Water at its southern one (Zenk et al., 1991; Martínez-Marrero et al., 2008).
Finally, the PUC is located below the CUC flowing northward on the continental slope (Barton, 1989; Machín and Pelegrí,
2009; Machín et al., 2010; Pelegrí and Peña-Izquierdo, 2015).

The mesoscale activity constitutes a second main feature in the area of interest, which might be even more energetic than the average flow itself (Sangrà et al., 2009). Three mesoscale domains may be defined: the Canary Eddy Corridor (CEC,
Sangrà et al. (2009)), the CVFZ and the upwelling front. The CEC is located downstream of the Canary Islands where the interaction between the southward flow and the archipelago generates long-lived eddies (Arístegui et al., 1994; Barton et al.,
1998; Sangrà et al., 2007, 2009; Ruiz et al., 2014; Barceló-Llull et al., 2017a). The second mesoscale domain is the CVFZ,
where several meanders and eddies produce strong interleaving between the water masses involved (Pérez-Rodríguez et al.,
2001; Martínez-Marrero et al., 2008). In this domain, the CC and the CUC separate from the African coast fueling the NEC,
giving rise to a shadow zone featured by poorly ventilated waters (Luyten et al., 1983). The third area is the front arising between the coastal upwelled waters and the stratified interior waters, defining the Eastern Boundary Upwelling System (EBUS)
in the Northwest African region (Mittelstaedt, 1983; Pastor et al., 2008; Arístegui et al., 2009). This EBUS is actually located off the African slope from the Gulf of Cadiz until Cape Blanc/Cape Verde in summer/winter with a high mesoscale vari-



ability in the form of both filaments and eddies (Hagen, 2001; Sangrà et al., 2009; Ruiz et al., 2014). The upwelling process raises nutrient-rich waters to the euphotic layer, developing a high primary production latitudinal band off Northwest Africa known as the Coastal Transition Zone (CTZ) (Barton et al., 1998; Pelegrí et al., 2006). The mesoscale activity plays an essen-

60 tial role as a lateral source of nutrients and organic matter towards the oligotrophic waters of the NASG (Barton et al., 1998; García-Muñoz et al., 2004; Pelegrí et al., 2006; Álvarez-Salgado et al., 2007; Sangrà et al., 2009).

The distribution of inorganic nutrients and organic matter in the ocean responds to a combined effect of physical and biogeochemical processes. Within the euphotic zone, primary production is solely limited by the availability of inorganic nutrients (IN) (Copin-Montegut and Copin-Montegut, 1983; Falkowski et al., 1998). Below the euphotic zone respiration generally ex-

65 ceeds primary production. As a result, the organic matter produced at the sea surface is remineralized in the subsurface layers and hence the concentration of IN increases from the interplay between the local rate of remineralization and the rate of water supply (Azam, 1998; Del Giorgio and Duarte, 2002; Pelegrí et al., 2006; Pelegrí and Benazzouz, 2015b).

The relevance of lateral advective transports on the spatio-temporal distribution of biogeochemical variables in the ocean has historically received little attention (Ganachaud, 1999; Ganachaud and Wunsch, 2002b). Specifically in the EBUS off North-

70 west Africa, some recent manuscripts related to lateral advective transports of biogeochemical variables have shed light on this topic (Álvarez and Álvarez-Salgado, 2009; Alonso-González et al., 2009; Santana-Falcón et al., 2017; Fernández-Castro et al., 2018).

The ocean dynamics in the region between $10°$ and $40°$N off Northwest Africa during two different seasons is addressed in this manuscript. An inverse box model is applied to hydrographic observations to estimate mass transports. This method

provides a velocity field consistent with both mass and properties conservation within a closed volume and with the thermal wind equation (Wunsch, 1996). Several authors have already described the circulation patterns of the NASG by applying an inverse model (Ganachaud and Wunsch, 2002a; Ganachaud, 2003b, a; Hernández-Guerra et al., 2005; Machín et al., 2006; Pérez-Hernández et al., 2013; Hernández-Guerra et al., 2017).

To sum up, the main goal of this manuscript is to estimate mass, nutrient and organic matter transports during fall and spring

south of the Canary Islands in the context of a highly variable environment as the CVFZ. The remaining of this manuscript is organized as follows: the dataset is presented in section 2; the seasonal distribution of the water masses and their properties is displayed in section 3; the technical details of the inverse box model are covered in section 4; the resulting velocity field and the corresponding mass, nutrient and organic matter transports are presented in section 5. Section 6 is devoted to the discussion to end up with some conclusions at section 7.

## 2 DATASET

COCA-I and COCA-II cruises were carried out in fall (10 September to 1 October 2002) and spring (21 May to 7 June 2003) respectively, aboard the BIO Hesperides as part of the research project Coastal-Ocean Carbon Exchange in the Canary Region (Hernández-León et al., 2019). The location of Conductivity-Temperature-Depth (CTD), inorganic nutrients (IN) and dissolved organic carbon (DOC) stations in COCA-I and COCA-II defines a closed box along three transects (Figure 1). The northern



transect (N) spans from station 1 to 32 at 26°N (section from stations 1 to 11 is tilted some 30° with respect to the east). The western transect (W) is located at 26°W from station 32 to 42. Finally, the southern zonal transect (S) at 21°N runs from station 42 to 63 (COCA-I) or 66 (COCA-II) over the continental slope (Table 1). The distance between neighbouring CTD stations was some 50 km except for the stations over the continental slope where this distance was shortened. Adjacent DOC and IN stations were separated by a variable distance, with its lowest value being about 50 km at stations closer to the coast.

CTD data were collected from the sea surface down to 2000 m depth with a vertical resolution of 2 dbar. Temperature was calibrated with 45 readings performed with a reversible digital thermometer, while salinity was calibrated by analysing 60 water samples with the Portasal salinometer. The residuals have an average value of $0.00013 \pm 0.00400$ °C and $0.0005 \pm 0.005$ in salinity.

Total organic carbon (TOC) was analyzed assuming that it is virtually in dissolved form. Water samples (10 mL) were
dispensed directly into glass ampoules, previously combusted at 500 °C during 12 h. 50 µL of $H_3PO_4$ were added immediately to the sample, sealed and stored at 4 °C until analyzed with a Shimadzu TOC-5000 (Sharp et al., 1993). Before the analysis, samples were sparged with $CO_2$-free air for several minutes to remove inorganic carbon. TOC concentrations were determined from standard curves (30 to 200 µM) of potassium hydrogen phthalate produced every day (Thomas et al., 1995). To check accuracy and precision, reference material from Jonathan H. Sharp laboratory (University of Delaware) was analyzed daily.
DOC distribution up to 2000 m depth presented a more representative coverage in fall than in spring (Fig. 2, green dots), despite in spring the number of stations was higher than in fall (Fig. 1, black circles; Tab. 1).

The three inorganic nutrient sampled were silicates ($SiO_2$), nitrates plus nitrites ($NO_x$), and phosphates ($PO_4$). These samples were frozen until measured with a Bran Luebe AA3 autoanalyser following the standard methodology established by Hansen and Koroleff (1999). Nutrient data covered up to 2000 m, while in fall they concentrated in the shallowest layers
($< 200$ m, Fig. 2, pink crosses).

Wind data were selected from the QuikSCAT database made available by CERSAT (Centre ERS d' Archivage et de Traitement, http://www.ifremer.fr/cersat/). These wind fields were averaged weekly with a spatial resolution of 0.5° (shown in Fig. 1 with half of the original spatial resolution). The Smith-Sandwell database with 1-minute horizontal resolution was used as the source of bathymetry data (Smith and Sandwell, 1997).

Freshwater flux data were estimated from the rates of evaporation and precipitation extracted from the Surface Marine Data 1994 of Da Silva (http://iridl.ldeo.columbia.edu/SOURCES/.DASILVA/.SMD94/). The climatological mean depths of the neutral density field for the years 2002 and 2003 were calculated from the climatological temperature and salinity extracted from the World Ocean Atlas 2013 (WOA13, https://www.nodc.noaa.gov/OC5/woa13/woa13data.html).

GLORYS (GLOBAL_REANALYSIS_PHY_001_025 product) issued by Copernicus Marine Environment Monitoring Ser-
vice (CMEMS, http://marine.copernicus.eu) was used as a primary source of dynamic variables. Its horizontal resolution is 1/12° with 50 standard depths. Hydrological data from GLORYS were also employed to diagnose the average oceanographic conditions during each cruise. This product assimilates field observations in real time.


SEALEVEL_GLO_PHY_L4_REP_OBSERVATIONS_008_047 product provided surface geostrophic currents estimated from sea level anomalies. These data capture the mesoscale structures and are helpful to validate the near-surface geostrophic
field estimated from the inverse model.

GLORYS-BIO (GLOBAL_REANALYSIS_BIO_001_029 product) produced daily mean 3D biogeochemical fields with the same resolution as GLORYS. This reanalysis forces the biogeochemical model with the nutrient initial conditions from WOA13. IN concentrations from GLORYS-BIO ($SiO_2$, $NO_3$, and $PO_4$) were used to assess nutrient transports by the model (in section 5).

The data treatment, the graphical representations and the inverse model are coded in MATLAB (MATLAB, 2018). The vertical sections are produced using the 'nearest' 2D interpolations, a method also employed in the estimates of the IN and DOC transports. Ocean Data View using the DIVA gridding method is employed to produce DOC concentration charts (Schlitzer, Reiner, 2019).

## 3  HYDROGRAPHY AND WATER MASSES

Neutral density $\gamma_n = \gamma_n(\theta, S, p)$ is used as the density reference variable, being the isoneutrals the surfaces where value of $\gamma_n$ is constant (Jackett and McDougall, 1997). The $\gamma_n$ vertical sections contain the surface (SW), central (CW), intermediate (IW) and deep water (DW) masses according to Macdonald (1998) for the North Atlantic at 24°N, represented with white dashed lines at 26.44, 27.38 and 27.82 $\mathrm{kg\,m^{-3}}$ (Figure 2). The x-axis direction is selected according to the path followed by the vessel during both cruises, starting in the northeast and finishing in the southeast of the domain. The N/W and W/S corners
are indicated with two vertical grey dashed lines at stations 32 and 42, respectively.

The $\Theta - S_A$ diagrams exhibit four regions delimited by potential density anomaly contours of 26.39, 27.30 and 27.72 $\mathrm{kg\,m^{-3}}$, equivalent to the isoneutrals which separate the main water masses (Fig. 3). These three isoneutrals are approximately at 132/123, 672/700 and 1294/1305 m depth (Fig. 2). The water masses sampled during both cruises are North Atlantic Central Water (NACW), South Atlantic Central Water (SACW), Antarctic Intermediate Water (AAIW), Mediterranean Water (MW),
and North Atlantic Deep Water (NADW) (Emery and Meincke, 1986; Macdonald, 1998; Emery, 2008). Their main hydrological characteristics are summarized in Table 2. Below the mixing layer and above 700 m ($26.44 < \gamma_n < 27.38\,\mathrm{kg\,m^{-3}}$), NACW and SACW are the dominant water masses. SACW is featured by a higher amount of nutrients, $1 - 2\,°C$ colder and $0.1 - 0.4$ fresher than NACW (Fig. 3 and Tab. 2). Below, from 700 up to 1300 m ($27.38 < \gamma_n < 27.82\,\mathrm{kg\,m^{-3}}$), the intermediate waters AAIW and MW are the dominant water masses (Hernández-Guerra et al., 2017). MW is a relatively warm and salty water
mass, while AAIW is colder and fresher (Tab. 2). Finally, below 1300 m ($\gamma_n > 27.82\,\mathrm{kg\,m^{-3}}$) the predominant water mass is NADW with *in situ* temperature and salinity values lower than $5.7\,°C$ and 35.14 (Tab. 2).

A description about the seasonal variability of the water masses is also performed with observations from the $\Theta - S_A$ diagrams (Fig. 3). The distribution of water masses is quite similar for both cruises. There is a higher temperature variability at surface waters during fall with maximum values 2-3 ºC higher than in spring. During spring, the variability observed at central
waters is associated to larger fluctuations in salinity affecting the whole water column. At DW there is a higher contribution of





NADW in the whole domain during fall. Finally, the surface layer is thicker in fall than in spring in all the sections made with respect to $\gamma_n$.

These seasonal differences may also be described transect to transect. The northern transect (Fig. 2, stations 2 to 32; Fig. 3, margenta dots) is occupied by NACW, AAIW, MW and NADW in both seasons. At intermediate levels, a higher contribution
of MW is observed in spring while a slightly higher contribution of AAIW is obtained in fall. The western transect (Fig. 2, stations 32 to 42; Fig. 3, dark grey dots) has a similar distribution as the northern one, with a lower variability in the upper layers and a smaller influence of MW. In the southern transect (Fig. 2, stations 42 to $63 - 66$; Fig. 3, blue dots), the highest spatio-temporal variability is observed. This variability at surface and central levels is associated to the position of the CVFZ and, in turn, to the meso- and submesoscale structures associated to the front. The CVFZ is located where the isohaline of 36,
or equivalently $S_A = 36.15 \, \mathrm{g \, kg^{-1}}$, intersects the $150 \, \mathrm{m}$ isobath (Zenk et al., 1991) (Fig. 4). CVFZ is found in the southern transect in its westernmost position in fall, at stations $46 - 48$. Hence, SACW with relatively low $S_A$ is observed above the upper limit of CW east of the CVFZ location (Fig. 4). In spring, the CVFZ shifts to a position closer to the African coast at station 52, with a water incursion of higher salinity NACW centred at station 58 (Figs. 4 and 5). At intermediate levels, MW is registered at the northern transect while in the southern one the predominant water mass is AAIW. Regarding the seasonal
variability, the contribution of MW in the northern transect is higher in spring while the contribution of AAIW in the southern transect is higher in fall.

Although the IN have been extracted from the model and the distributions of $\Theta$, $S_A$ and $\gamma_n$ have been obtained from the hydrographic data, there is a good agreement between the structures described by both datasets. The *in situ* concentrations of $\mathrm{SiO_2}$, $\mathrm{NO_X}$ and $\mathrm{PO_4}$ up to $250 \, \mathrm{m}$ depth (black dots in Fig. 6) are represented together with the time-averaged concentrations
of $\mathrm{SiO_2}$, $\mathrm{NO_3}$ and $\mathrm{PO_4}$ up to $2000 \, \mathrm{m}$ depth selected from GLROYS-BIO. In this way the IN outputs from the model are compared with *in situ* observations since their concentration in both cases present an acceptable match with the exception of $\mathrm{NO_X}$ and $\mathrm{PO_4}$ concentrations at the S transect. On the other hand, the IN model outputs look alike IN from historical *in situ* databases (not shown here).

At central levels, high IN concentrations have been sampled near the continental slope in both the northern (stations 10 to 18)
and southern (50 to 56) transects in fall. Values observed are 1-5 $\mathrm{\mu mol \, kg^{-1}}$ for $\mathrm{NO_3}$ and 0.1-0.4 $\mathrm{\mu mol \, kg^{-1}}$ for $\mathrm{PO_4}$ higher than those recorded in spring at similar places (Fig. 7). This might be related to long-lived mesoscale eddies or instabilities related to the CVFZ (Zenk et al., 1991; Sangrà et al., 2009). IN concentrations are notably high at intermediate and deep levels as compared to those at central levels (Fig. 6) and have the same order of magnitude as those documented before in the domain (Pérez et al., 2001; Pérez-Hernández et al., 2013). The distributions of $\mathrm{SiO_2}$, $\mathrm{NO_3}$ and $\mathrm{PO_4}$ are similar in both cruises and
their concentrations increase with depth as a result of the remineralization of organic matter (Fig. 7). The area where the least nutrients are found at depth throughout the domain is the northwest corner of the box (stations 24 to 32). With respect to the IN seasonal variability at intermediate depths, the three concentrations do not present large differences between the values measured in fall and spring (Figs. 7 and 3). In both seasons the concentrations of $\mathrm{SiO_2}$, $\mathrm{NO_3}$ and $\mathrm{PO_4}$ are 4-6, 2-6 and 0.2-0.4 $\mathrm{\mu mol \, kg^{-1}}$ higher in AAIW than in MW (Tab. 2). The NADW is characterized by a moderate increase of $\mathrm{SiO_2}$ and by
a slight decrease of $\mathrm{NO_3}$ and $\mathrm{PO_4}$ with respect to the values documented here at intermediate levels. In both seasons, the



maximum concentrations of $SiO_2$ are 28-29 $\mu mol\,kg^{-1}$. Nevertheless, and specifically in spring, maximum concentrations of $NO_3$ and $PO_4$, 28 $\mu mol\,kg^{-1}$ and 1.8-1.9 $\mu mol\,kg^{-1}$, are lower than those recorded at intermediate levels, providing a similar vertical variability as that reported by Machín et al. (2006) (Tab. 2).

DOC concentrations are higher and more widely distributed in the water column in fall than in spring, when the DOC maximum values are more confined to surface and central waters (Figs. 8 and 6, Tab. 2). This fact is especially significant in the southern transect occupied by SACW (Fig. 6). This last water mass presents maximum concentrations of DOC $35 - 40$ $\mu mol\,L^{-1}$ lower than those found for NACW (Tab. 2). This difference is more pronounced in spring season (Tab. 2). In addition, the fall DOC observations present a larger variability in central waters as previously seen for IN. Lower DOC concentrations are observed for stations sampled in the western transect while the highest concentrations are recorded in the stations next to the African slope with values above 100 $\mu mol\,L^{-1}$ (Fig. 8). On the other hand, it is noteworthy the high concentrations of DOC recorded at intermediate waters of the northern transect in both cruises (Figs. 8 and  6).

## 4   THE INVERSE MODEL

An inverse box model is applied to the hydrographic data of the two COCA cruises to provide the absolute velocity field across the three sections (Wunsch, 1978). This method has been widely applied in different areas of the Atlantic Ocean as an efficient method to obtain absolute geostrophic flows (Martel and Wunsch, 1993; Paillet and Mercier, 1997; Ganachaud, 2003a; Machín et al., 2006; Pérez-Hernández et al., 2013; Hernández-Guerra et al., 2017; Fu et al., 2018). Assuming geostrophy and the conservation of mass and other properties in the ocean bounded by the African coast and the hydrological sections, the velocity fields are obtained allowing an adjustment of freshwater flux and Ekman transports.

### 4.1   Selection of layers

The closed ocean where the inverse model is applied is divided into nine layers by means of the neutral densities defined by Macdonald (1998) and modified by Ganachaud (2003a) for the North Atlantic Ocean. This distribution is slightly modified to include two layers instead of one between 26.85 and 27.162 $kg\,m^{-3}$ by adding the isoneutral 27.035 $kg\,m^{-3}$ as others authors have done previously in this side of the NASG (Comas-Rodríguez et al., 2011; Pérez-Hernández et al., 2013). The location of the isoneutrals are represented in Figure 2. The upper five layers group the surface and central waters, with the first layer until the isoneutral 26.44 $kg\,m^{-3}$ being related to surface waters and the 4 remaining layers between 26.44 $kg\,m^{-3}$ and 27.38 $kg\,m^{-3}$ to central waters. The intermediate waters are found in the next two layers between 27.38 and 27.82 $kg\,m^{-3}$ while the deepest two layers below 27.82 $kg\,m^{-3}$ contain the upper deep waters.

### 4.2   The system of equations

The inverse box model takes into account mass conservation per layer and also in the whole water column. The salinity is actually introduced as a salinity anomaly, which is also conservative within individual layers and in the whole water column (Ganachaud, 2003b). On the other hand, heat is introduced as a heat anomaly in the two deepest layers where it is also





considered conservative. The salinity and heat are added as anomalies to improve the conditioning of the inverse model and get a higher rank in the system of equations by reducing the linear dependency between equations (Ganachaud, 2003b).

Therefore, the model is composed of a set of 22 equations (10 for mass conservation, 10 for salt anomaly conservation and
2 for heat anomaly conservation). Those equations are solved for 32 and 34 unknowns, comprised of 28/30 reference level velocities in fall/spring, 3 unknowns for the Ekman transport adjustments (one unknown per section), and 1 unknown for the freshwater flux. The resulting system is undetermined and a Gauss-Markov estimator is used to select a solution by adding *a priori* information. This *a priori* information consists of the uncertainties for both the unknowns ($R_{xx}$) and the noise of the equations ($R_{nn}$).

### 4.2.1    Uncertainties of unknowns ($R_{xx}$)

The geostrophic velocity field is calculated in the central position between two consecutive stations. The isoneutral selected as the reference level is the deepest common $\gamma_n$ for all the stations, $27.962 \, \mathrm{kg \, m^{-3}}$ (Fig. 2). The variance of the velocity in the reference level at each location is used as a measure of the *a priori* information. These variances are calculated with an annual mean velocity extracted from the daily velocity provided by GLORYS. These velocities are interpolated to the reference level
depth. This reference level depth is estimated from the climatological mean depth of $27.962 \, \mathrm{kg \, m^{-3}}$ extracted from WOA13. The stations closer to the coast in the northern and southern transects have the highest variability in the velocity field.

The initial Ekman transports are estimated from the wind stress for both cruises. The uncertainty associated to these Ekman transports is related to the error in their measurements and to the variability of the wind stress. A 50% uncertainty is assigned to the initial estimate of Ekman transports. The initial freshwater flux is a climatological mean of 0.0171 Sv, which is also assigned
an uncertainty of 50 % as reported in similar approaches (Ganachaud, 1999; Hernández-Guerra et al., 2005; Machín et al., 2006).

Both the Ekman transports and freshwater flux with their uncertainties are added to the model in the conservation equations corresponding to the shallowest layer of the mass transport and salt anomaly and also in the conservation equations of total mass transport and total salt anomaly.

### 4.2.2    Uncertainties in the noise of equations ($R_{nn}$)

The noise of each equation depends on the density field, on the layer thickness and on the uncertainties of the unknowns (Ganachaud, 1999, 2003b; Machín et al., 2006). In fact, Ganachaud (2003b) established that the largest source of uncertainty in conservation equations arises from the deviation of the baroclinic mass transport from their mean value at the time of the cruise. Thus, an analysis of the annual variability in the velocity field for the nine layers is performed. The velocity variability
is examined in the mean depth between two successive isoneutral surfaces whose climatological mean depths are defined by WOA13. These variabilities are included in the inverse model as the uncertainties of the *a priori* noise of equations in terms of variances of mass, salt anomaly and heat anomaly transports. The velocity variance from the annual mean velocity for each layer are estimated with GLORYS and transformed into transport values by multiplying times density and the vertical area of





the section involved. These *a priori* transport uncertainties are presented in Table 3. Furthermore, the uncertainty assigned to
the conservation equation in the total mass is the sum of the uncertainties from the rest of the nine conservative mass equations.

The equations for salt and heat anomaly conservation depend on both the uncertainty of the mass transport and the variance of these properties (Ganachaud, 1999). In these cases, the *a priori* noise of each equation will not depend strictly on the water mass but on the layer considered, as shown in the following equation (Ganachaud, 1999; Machín, 2003):

$$R_{nn}(Cq) = a * var(C_q) * R_{nn}(mass(q)) \tag{1}$$

where $R_{nn}(Cq)$ is the uncertainty in the anomaly equation of the property (salt or heat anomaly); $var(C_q)$ is the variance
of this property; $a$ is a weighting factor of $4$ in the heat anomaly, $1000$ in the salt anomaly and $10^6$ in the total salt anomaly; $q$ is a given equation corresponding to a given layer.

## 5   RESULTS

### 5.1   Velocity fields and mass transports

Figure 9 shows the reference level velocities obtained after the inversion is performed. The variance of these velocities are
also estimated by the model. The uncertainties are much higher than the values themselves and around $\pm$ (0.5-1) $\mathrm{cm\,s^{-1}}$.
During fall all non-zero values are positive, while in spring they are negative. This difference is important mainly in the
western and southern transects where the module of the velocity increases reaching values of $0.3$ and $-0.16$ $\mathrm{cm\,s^{-1}}$ in fall
and spring, respectively. Furthermore, the estimated reference level velocity values in the northern transect in spring are too
small, $O(10^{-4} - 10^{-5})$, while they take positive and significant values between $0.13$ and $0.25$ $\mathrm{cm\,s^{-1}}$ in some locations of this
transect in fall.

Once the geostrophic velocities at the reference level are estimated, they are integrated into the entire water column obtaining
the absolute geostrophic velocities (Fig. 10). These results are validated by comparison with the surface geostrophic velocity
and the sea level anomaly (SLA) derived from altimetry during both cruises (Fig. 11). To do this, the average fields of SLA and
geostrophic velocity at the sea surface are calculated and shown as a representation of the synoptic situation during both sur-
veys. Furthermore, the mass transports at the shallowest layer (red bars in Fig. 11), are superimposed with the aim of comparing
these transports with the average velocity field from altimetry. A remarkable mesoscale activity can be identified at both the
absolute geostrophic velocity sections (Fig. 10) and at the temporal average of SLA and the geostrophic velocity (Fig. 11). In
this last case, the structures are somewhat displaced with respect to their positions in the *in situ* velocity sections. For instance,
an anticyclonic eddy is located between stations 10 and 16 in the N transect in both seasons. This eddy, observed in autumn
with high velocities at intermediate layers, weakens in spring. This mesoscale structure could be part of the CEC (Sangrà et al.,
2009). Furthermore, it coincides with the position of an anticyclonic eddy previously documented (Barceló-Llull et al., 2017a;
Barceló-Llull et al., 2017b; Estrada-Allis et al., 2019).

In fall, two eddies are linked in the S transect, an anticyclonic one between stations 48 and 52 and a cyclonic one between
stations 52 and 60, both associated with the CVFZ. In spring, two anticyclonic eddies are observed, one centred at station





36 and the other one at station 56 also associated with CVFZ. In both seasons, mesoscale structures present a large vertical extension (Figure 10). In fall, these structures have higher velocities at IW and DW levels and they also affect a higher extension along each transect. In spring instead, these structures are vertically shortened (Fig. 10). The SLA also shows a high variability region with more intense structures in fall than in spring (Fig. 11).

Mesoscale structures are also visible in the vertical sections of $NO_3$ and $PO_4$ in fall, when their concentrations are higher than those observed in spring at similar locations (Fig. 7). Furthermore, high concentrations of DOC in fall at CW levels are recorded in the same area where the deep anticyclonic eddy is located, between stations 8 and 18 (Fig. 8). In spring, mesoscale structures in the vertical sections of IN and DOC at CW levels are less intense than in fall (Fig. 10). Nonetheless, DOC concentrations below the two anticyclonic structures at CW levels in spring are higher than at their surroundings.

The accumulated geostrophic mass transport is integrated to group the variability at different levels, having the first shallowest layer for SW, the next four layers for CW, then two layers for IW and the deepest two layers for DW (Figure 12). The total accumulated geostrophic mass transport, integrated for all the nine layers, is also represented. The horizontal axis has the same direction as the rest of the vertical sections and the three transects are separated by two vertical dashed grey lines. Sv is used here as equivalent to $10^9 \, \mathrm{kg \, s^{-1}}$. The positive/negative transport values indicate outward/inward transports from/to the box.

The accumulated mass transports show a significant horizontal spatial variability, especially marked in the southern transect in accordance to the geostrophic velocity distribution (Fig. 10). The presence of significant mesoscale structures might be one of the sources for the total imbalances in the accumulated mass transport. In fall, the total imbalance is -1.43 Sv and in spring 3.55 Sv (Tab. 4).

On the other hand, the geostrophic mass transport can be integrated per layer and transect together with the total imbalance

inside the box and the total mass transport uncertainty per layer (black line and horizontal black bars in Fig. 13). Moreover, Table 4 compiles these transports integrated for the different water levels. More than 65% of the mass transport is given at SW and CW levels (Tab. 4). In fall, these water masses mostly get into the box across the northern and southern transects with transports of $-5.61 \pm 1.86$ Sv and $-4.35 \pm 1.48$ Sv, respectively; the mass leaves the box by flowing westward with a value of $5.96 \pm 1.75$ Sv. In spring, water masses also get in the box mostly through the northern transect with $-6.69 \pm 1.63$ Sv but

they set off along the western and southern transects with transports of $4.05 \pm 1.75$ Sv and $5.20 \pm 1.55$ Sv, respectively. It is remarkable how the inward transport in fall across the southern transect is reversed to an outward flow in spring at the southern transect (Fig. 13).

The position of CVFZ in both seasons could partly explain that seasonal variability in the mass transports at central levels (Fig. 14). In fall, the CVFZ is located further from the African coast, so SACW is present at almost all stations of the south

transect. This location of the CVFZ prevents a latitudinal mass transport from north to south. However, in spring the CVFZ is closer to the African slope allowing an important mass transport from north to south.

Between 5 and 30% of the mass transport is given in intermediate levels (Tab. 4). In fall, the intermediate water transport directs northward in the southern transect with $-1.93 \pm 1.69$ Sv and it leaves the box with $1.94 \pm 1.85$ Sv and $0.48 \pm 1.71$ Sv across the northern and western transects, respectively. During spring, this transport weakens and changes its direction in the





northern and southern transects with transports of $-0.48 \pm 1.65$ Sv and $0.39 \pm 1.73$ Sv, respectively, increasing its westward
transport to $1.21 \pm 1.68$ Sv.

The mass transport in deep water layers barely exceeds 3% (Tab. 4). An exception is the 8% given in the northern transect
during fall where the estimated transport is $0.73 \pm 1.71$ Sv. In both cruises the transport at deep levels is nearly balanced.

## 5.2 Nutrient and DOC transports

DOC and IN transports are obtained by multiplying their concentration times mass transports. DOC, IN and geostrophic
velocities are obtained at different grids, so they need to be interpolated to the same grid. In the case of DOC, the velocities
are interpolated to the points where the concentrations of DOC are taken and, in a second step, the concentrations of DOC are
linearly interpolated to the depths of the geostrophic velocities. On the other hand, the *in situ* measurements of IN are scarce
at IW and DW where their concentrations become important. Therefore, instead of using the observational data, the average
outputs of GLORYS-BIO are used to estimate the IN transports. $SiO_2$, $NO_3$, and $PO_4$ mean concentrations are interpolated to
the grid nodes where the geostrophic velocities are estimated by the inverse model.

DOC transports are obtained by subtracting a refractory concentration of $40$ µmol $L^{-1}$ from the measured DOC as other
authors do (e.g., Santana-Falcón et al., 2017). This is done because the refractory fraction renewal is thousands of years, a
period much longer than the time required in the processes we are focused on (Hansell, 2002). On the other hand, it should be
emphasized that DOC transports may be underestimated due to the scarcity of measurements performed.

The IN transport values are being presented in the text always ordered as $SiO_2$, $NO_3$ and $PO_4$ (Figures 15 and 16). Tables 5,
6 and 7 summarize those transports integrated per water level and transect. The errors are relative to the mass transport errors
and are calculated as the standard deviations of IN transports. On the other hand, the DOC transport estimates per layer and
transect are also shown in Figure 16 and summarized per water level and transect with their relative error (calculated as in the
IN transports) in Table 8. In order to be able to compare our transport values of IN and DOC with those reported by other
authors, units of $kmol\,s^{-1}$ and $\times 10^8\,mol\,C\,day^{-1}$ are employed for IN and DOC transports, respectively, being both units
equivalent.

IN enter the domain both from north and south at CW in fall. At the northern transect the transports are relatively low while at
the southern one transports double the amount coming from north, with $-0.41 \pm 0.11$, $-0.78 \pm 0.21$ and $-0.05 \pm 0.01\,kmol\,s^{-1}$.
In spring, instead, the IN transports change their direction in the southern transect and only enter from the north with values
which double those during fall, $-0.40 \pm 0.09$, $-0.90 \pm 0.21$, $-0.06 \pm 0.01\,kmol\,s^{-1}$. On the other hand, IN transports at CW
layers are overall westward with low values in fall while in spring IN transports are southward and westward.

At IW levels, during fall the IN transports are inward through the southern transect with $-0.27 \pm 0.24$, $-0.36 \pm 0.32$,
and $-0.02 \pm 0.02\,kmol\,s^{-1}$, and to a lesser extent through the western transect. Outward transports are observed through
the northern transect with $0.23 \pm 0.22$, $0.30 \pm 0.28$ and $0.02 \pm 0.02\,kmol\,s^{-1}$. In spring, the IN enter weakly through the
northern transect and leave the box crossing the western and southern transects with significant values of $0.19 \pm 0.27$ and
$0.12 \pm 0.55\,kmol\,s^{-1}$ for $SiO_2$; $0.25 \pm 0.35$ and $0.17 \pm 0.75\,kmol\,s^{-1}$ for $NO_3$; and $0.02 \pm 0.02$ and $0.01 \pm 0.05\,kmol\,s^{-1}$





for $PO_4$. In summary, while in fall the main IN transports are in the south to north direction, in spring they are mainly
southwestward like the mass transport behaviour at these levels during this season (Tab. 4).

Finally, at DW during both seasons, the net transports of the three nutrients are similar to those at IW but with smaller values
due to the low velocities at these depths, despite their high nutrient concentrations (Figs. 15 and 16). Furthermore, the relative
error in these layers is always larger than the IN transport values.

In spring, DOC transports at SW and CW levels are the same order of magnitude and one order of magnitude higher than
those at IW levels. In turn, these transports at IW levels are one order of magnitude higher than those at DW levels during this
season. In contrast, during fall at the northern transect DOC transports have the same magnitude in both SW, CW and IW and
they are one order of magnitude smaller than those at CW levels during spring (Tab. 8). In this season, DOC transports at SW
and CW of the western transect have unrealistic small values likely related to the low amount of measurements made in this
transect during fall. DOC transports through the northern transect could also be somewhat underestimated for the same reason.
However, at the southern transect during fall, the result is of the same order of magnitude as in spring.

In spring, DOC transports behave in a similar way in all the water column. At SW and CW levels, $-2.33\pm0.57 \times10^8\,\mathrm{mol\,C\,day}^{-1}$
enter through the northern transect, of which $0.89\pm0.25 \times10^8\,\mathrm{mol\,C\,day}^{-1}$ leave the box through the southern transect and ap-
proximately a half of it through the western transect. During fall, there is an important outward DOC transport at SW, CW and
IW levels, specially southward through the southern transect at SW and CW levels with a total of $1.48\pm0.66 \times10^8\,\mathrm{mol\,C\,day}^{-1}$
(Tab. 8).

Two opposite trends can be observed when both cruises are compared. In fall the IN net transports are $-0.34\pm0.20$, $-0.67\pm$
$0.40$ and $-0.04\pm0.02$ kmol s$^{-1}$ at CW levels; $-0.17\pm1.07$, $-0.23\pm1.39$ and $-0.01\pm0.09$ kmol s$^{-1}$ at IW levels, and
$-0.12\pm0.25$, $-0.10\pm0.21$ and $-0.01\pm0.01$ kmol s$^{-1}$ at DW levels. The amount of nutrients entering the box is larger than
those leaving the box with the exception at the shallowest level where the IN leave the box (Tabs. 5, 6 and 7 and Figs. 15 and 16).
On the other hand, the net DOC transports are outward for both SW, CW and IW levels with $0.10\pm0.13 \times10^8\,\mathrm{mol\,C\,day}^{-1}$
at SW level, $1.34\pm0.80 \times10^8\,\mathrm{mol\,C\,day}^{-1}$ at CW levels, and $0.12\pm0.72 \times10^8\,\mathrm{mol\,C\,day}^{-1}$ at IW (Tab. 8 and Fig. 16).

In contrast, during spring a net outward transport is obtained for the three IN with $0.28\pm0.61$, $0.57\pm1.22$ and $0.04\pm$
$0.08$ kmol s$^{-1}$ at CW, $0.28\pm0.72$, $0.36\pm0.94$ and $0.02\pm0.06$ kmol s$^{-1}$ at IW, and $0.13\pm6.79$, $0.12\pm6.26$ and $0.01\pm$
$0.42$ kmol s$^{-1}$ at DW (Tabs. 5, 6 and 7, and Figs. 15 and 16). On the other hand, the DOC net transports are inward with
$-0.14\pm0.08 \times10^8\,\mathrm{mol\,C\,day}^{-1}$ at SW level; $-0.80\pm1.72 \times10^8\,\mathrm{mol\,C\,day}^{-1}$ at CW levels; and $-0.01\pm0.02 \times10^8\,\mathrm{mol\,C\,day}^{-1}$
at IW levels (Tab. 8 and Fig. 16).

## 6   DISCUSSION

The circulation patterns in the studied area of the Canary Basin change significantly showing a seasonal variability from fall
to spring. The differences between the two seasons are reflected in the estimated mass transports for both cruises (Fig. 13 and
Tab. 4).





Trade Winds are intense all year long between the Canary Islands and Cape Blanc (26° N to 21° N), and generate a quasi-permanent upwelling in this region north of Cape Blanc. In contrast, the developed EBUS intensity and its off-shore development change from fall to spring (Benazzouz et al., 2014). In the beginning of spring there is a strong heating that generates a sharp water stratification particularly in the interior ocean of the NASG and a very intense upwelling which makes the EBUS to develop strongly far off-shore. In early fall, the EBUS weakens and becomes a shallower front which approaches towards

the coast (Pelegrí and Benazzouz, 2015a). In fact, the variability related to its location and intensity may be the cause that the estimated mass transports in the north-south direction are distributed between levels of central waters and intermediate waters in fall, and that in spring these mass transports parallel to the coast are confined to the shallowest layers at central waters. On the other hand, these changes in the EBUS and in the water stratification may also be related to the westward mass transports which in fall are accentuated and confined to the levels of SW and CW, as a shallow Ekman transport, while in spring the

lateral westward transport is distributed from the sea surface down to IW levels (Tab. 4 and Fig. 13).

SW transports through the N and W transects show similar patterns but in fall they are significantly more intense than in spring. In addition, CW level transports through these two transects show also similar patterns with a low variability between both seasons. The largest differences are observed in the estimated transports through the S transect which changes from fall to spring, where the transport is northward during fall and southward during spring. This observed variability in the transports in

SW and CW levels in the southern part of the domain is likely related to the seasonal changes in the position of CVFZ which in turn is related to the seasonal changes in the North Atlantic Tropical Gyre (NATG), south of the domain (Pelegrí et al., 2017). The fact that the Intertropical Convergence Zone moves southward in winter and northward in summer affects the circulation patterns south and north of Cape Blanc (Lázaro et al., 2005; Stramma et al., 2008; Peña-Izquierdo et al., 2012). While in fall the CVFZ crosses the S transect in its westernmost position, in spring it moves closer to the African coast. The output of the

GLORYS model matches the observations during both seasons (Fig. 14). In addition, the dynamics described by the geostrophic field of GLORYS also agree with the velocity field and the mass transports at CW levels estimated by the inverse model in the S transect for both seasons.

GLORYS velocity outputs also reproduces the meso and submesocale associated with the CVFZ (Pérez-Rodríguez et al., 2001; Martínez-Marrero et al., 2008) which are observed directly in the S transect of the velocity sections (Fig. 10) and in the

accumulative mass transport (black line in Fig. 13). Specifically during fall, the reported eddies boost a significant transport at SW and CW levels from south to north. All these results at CW levels are consistent with the late-summer and fall growth of the Mauritania Current and of the PUC and also with the decrease of the NATG currents and the weakening of the Guinea Dome in winter and spring seasons (Siedler et al., 1992; Lázaro et al., 2005; Peña-Izquierdo et al., 2012; Pelegrí and Peña-Izquierdo, 2015; Pelegrí and Benazzouz, 2015a). The estimated transports at IW also show seasonal changes between fall and spring. This

region is featured by a late summer northward progression of AAIW observed in fall, and by a weak southward flow of MW in spring (Machín et al., 2010).

In general, the estimated transports of the three IN show similar patterns, very marked by the mass transport variability during both seasons. The level with the highest transport in all the nutrients at both seasons is the deepest CW layer.





This is quite in agreement with the local maximum of remineralization found for all tracers in the upper intermediate layer
Fernández-Castro et al. (2018).

CW levels are featured by a relatively high biological production and therefore a nutrient deficit, and also by large geostrophic velocities. During fall the amount of IN that enters the box through N and S transects is larger than the IN quantity that leaves the box through the W transect. In spring, on the other hand, the amount of IN transported outward through the W and S transects is larger than the IN which enters from the north.

At IW levels the concentrations of IN are high and stable related to the dominant remineralization process. During spring, the spatial distribution of the three IN transports are the same as at CW levels with smaller values. In this season the transports of IN are directed westward through the W transect towards the oligotrophic open ocean. In fall, the IN transports at IW levels have a behaviour different than at CW levels being the main transport in the south-north direction.

The most significant differences between the DOC transports in fall and spring are obtained in the first and second shallowest
layers where there are high lateral velocities and where the euphotic layer is located. During fall, the DOC quantity that enters by the north transect is a third of the amount that leaves the region by the south. In the spring, however, the large amount of DOC that enters the domain from the north doubles the quantity that leaves it by the S transect while a quarter does by the western transect.

In spring, when the stratification is less marked, the most significant and deepest transports of IN are observed toward
the open ocean in central and intermediate water levels. However, in fall, when the water column is more stratified and the upwelling process is the main physical forcing for nutrient supply at CW levels (Pastor et al., 2013), the IN transports toward oligotrophic interior ocean is less than in spring. In fact, while in the western transect during spring the IN transports increase with depth to their maximum values at the deepest central layer, in fall the opposite occurs, since the westward IN transports decrease with depth until cancelling at the last central layer; these transports reverse towards the coast at the two intermediate
layers (green line in Figs. 15 and 16).

On the other hand, DOC transports are deeper and more intensified toward the open ocean during spring than in fall. Nonetheless, in fall there is an important and deeper transport of IN in a direction parallel to the coast. In fact, at IW DOC concentrations accumulate next to the African coast in the upwelling region. Furthermore, inside the upwelling region at the N and S transects in fall, the two observed mesoscale anticyclonic eddies could accentuate this process.

The variability in intensity of the stratification, strength of upwelling and the position of the boundary between the upwelling and the oligotrophic interior ocean together with important meso and submesocale structures control the nutrients availability at CW and IW waters. It is also deduced from DOC transport estimates that the upwelling drives the changes in the size of the high production domain and equivalently, the position for the eastern boundary of the oligotrophic region in this area (Pastor et al., 2013).

The estimated transports of IN and DOC tell us that in fall there is a pronounced import of IN into the domain (with the exception of the SW layer) and a moderate export of DOC, especially at CW and IW levels. On the other hand, during spring there is a pronounced export of IN from the domain at CW and IW levels and a slight import of DOC at the shallowest CW levels and at the SW layer.





# 7 CONCLUSIONS

In summary, the net mass transports across the 3 transects are shown in Figure 17 for the different water layers and for both seasons. In this figure, the main changes of the net mass transports from fall to spring are observed. The net mass transport at CW layers coincides in both seasons in the N transect with a southward flow of $-2.94 \pm 1.26$ Sv in fall that increases in spring to $-4.89 \pm 1.14$ Sv. In the W transect the net westward mass transport at CW levels weakens from a value of $3.50 \pm 1.09$ Sv in fall to $2.96 \pm 1.06$ Sv in spring. The most remarkable change in the net mass transport at CW layers occurs in the southern

transect where in fall the net mass transport is northward with a value of $-3.85 \pm 1.03$ Sv, while in spring it is southward with a value of $2.80 \pm 1.02$ Sv. In the shallowest layer, the SW transports follow a pattern similar to those at CW levels across the 3 transects. However, in fall higher SW transports through the N and W transects are obtained, $-2.67 \pm 0.60$ and $2.46 \pm 0.66$ Sv, respectively, and in spring there is a higher SW transport through the S transect with $2.40 \pm 0.53$ Sv.

   At IW layers, the net transport in the south-north direction is intense and northward in fall, $1.94 \pm 1.85$ Sv, while it weakens

and reverses southward in spring, $-0.48 \pm 1.65$ Sv. In the W transect, the net westward mass transport at IW layers is less intense in fall, $0.48 \pm 1.71$ Sv, than in spring, $1.21 \pm 1.68$ Sv. Last, the net mass transport at DW levels is small as compared to the other water levels, with the exception of the $0.73 \pm 1.71$ Sv estimated in the N transect during fall.

   From a physical perspective, it has been analysed how these transports influence the transports of IN and DOC in the study area. It has also been observed how other non-physical factors determine the distribution of IN and DOC in the area of interest

due to their link within the biogeochemical cycles. It is noted how in fall, the domain works as a nutrient sink with total IN net import of $-0.61 \pm 1.97$, $-0.74 \pm 2.40$ and $-0.05 \pm 0.15$ kmol s$^{-1}$ of $SiO_2$, $NO_3$ and $PO_4$, respectively, and how in spring it works as a source of nutrients with a total nutrient net export of $0.73 \pm 0.91$, $1.21 \pm 1.51$ and $0.08 \pm 0.1$ kmol s$^{-1}$. And it is also observed how the net DOC outward transport is of $1.55 \pm 5.01 \times 10^8$ mol C day$^{-1}$ in fall makes the domain act as a source of DOC in this season and how the significant negative value of $-0.95 \pm 1.19 \times 10^8$ mol C day$^{-1}$ describes it as a DOC sink in

spring.

   However, what is really interesting here is to analyse the lateral transports of both IN and DOC towards the oligotrophic ocean and their seasonal variability. During spring there is a continuous westward IN transport, $0.75 \pm 0.37$, $1.34 \pm 0.66$ and $0.08 \pm 0.04$ kmol s$^{-1}$ of $SiO_2$, $NO_3$ and $PO_4$, respectively, toward the open ocean through the W transect in all the water column that coincide with an important transport of DOC, $0.50 \pm 0.25 \times 10^8$ mol C day$^{-1}$, mainly at SW and CW. In fall,

these transports are weakened at CW and reverse at IW, which means that the net westward transport of IN is smaller than in spring, with values of $0.03 \pm 0.01$, $0.35 \pm 0.13$ and $0.02 \pm 0.01$ kmol s$^{-1}$ for $SiO_2$, $NO_3$ and $PO_4$ towards oligotrophic waters. Westward transport of DOC is not observed even at the shallowest layer.

   It is still necessary to continue with the understanding of the physical and biogeochemical processes and the interactions between the productive EBUS and the interior ocean in its vicinity, especially in dynamically complex regions as this area

where the EBUS interacts with the CVFZ. Larger and more robust hydrological and biogeochemical databases would help to achieve this goal.





*Acknowledgements.* This work has been done thanks to the project COCA (REN2000-U471-CO2-02-MAR) and it was supported by the project FLUXES (CTM2015-69392-$C$3-3-R), both of them funded by the Spanish National Research Program. Currently, NB is working on her Ph.D. with a fellowship funded by the Spanish Ministry of Economy and Competitiveness.





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





**Figures and tables**

**Figure 1.** Hydrological (red dots), inorganic nutrients (pink circles) and DOC (black circles) sampling stations during cruises COCA-I (top) and COCA-II (bottom). Time-averaged wind stress during each cruise is also represented with the inset arrow denoting the scale (shown with half of the original spatial resolution).



**Figure 2.** $\gamma_n$ vertical sections during fall (top) and spring (bottom) cruises. White dashed isoneutrals limit the different water type layers. The direction chosen for the representation of the transects is the course of the vessel. Distance is calculated with respect to the first station (2). The section is divided into three transects: northern transect from east to west (from station 2 to 32), western transect from north to south (from station 32 to 42) and southern transect from west to east (from stations 42 to 63/66). The 3 transects are separated by two vertical grey dashed lines located at stations number 32 and 42. The sampling points of IN and DOC used in this work are also represented in pink crosses and green dots, respectively.

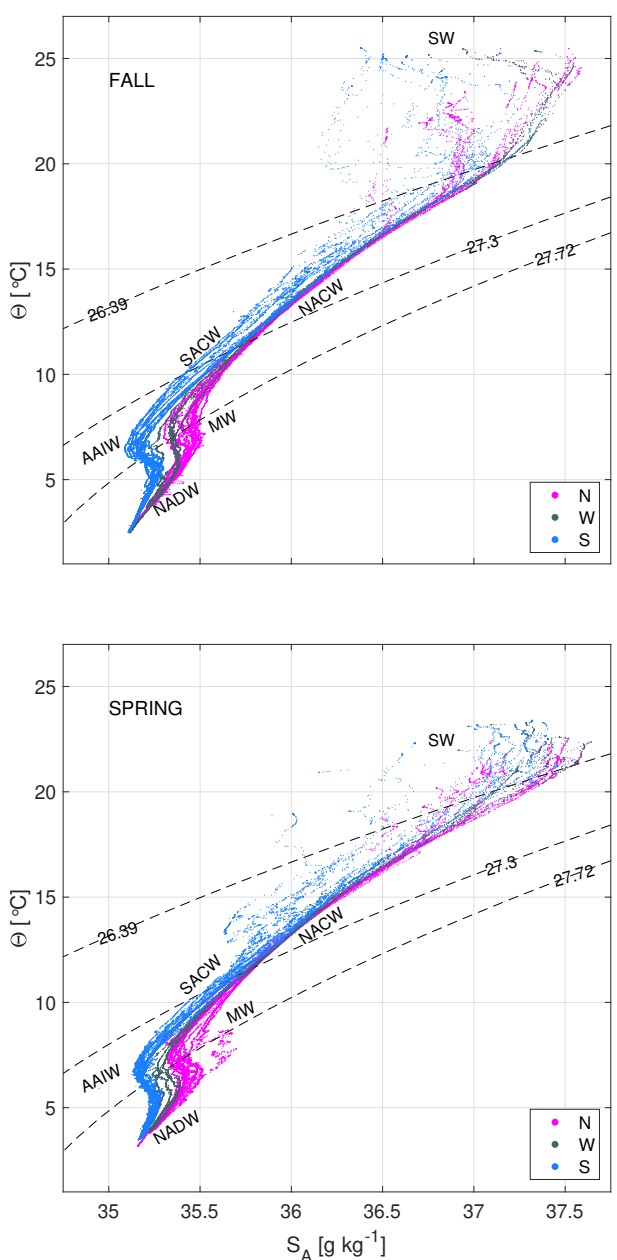

**Figure 3.** $\Theta - S_A$ diagrams of the hydrological measurements in fall (top) and spring (bottom) cruises. The different water masses at north (N, magenta dots), west (W, dark grey dots) and south (S, blue dots) transects are SW, NACW, SACW, AAIW, MW and NADW. Potential density anomaly contours equivalent to $26.44$, $27.38$ and $27.82$ kg m$^{-3}$ isoneutrals delimit the surface, central, intermediate and deep water levels.



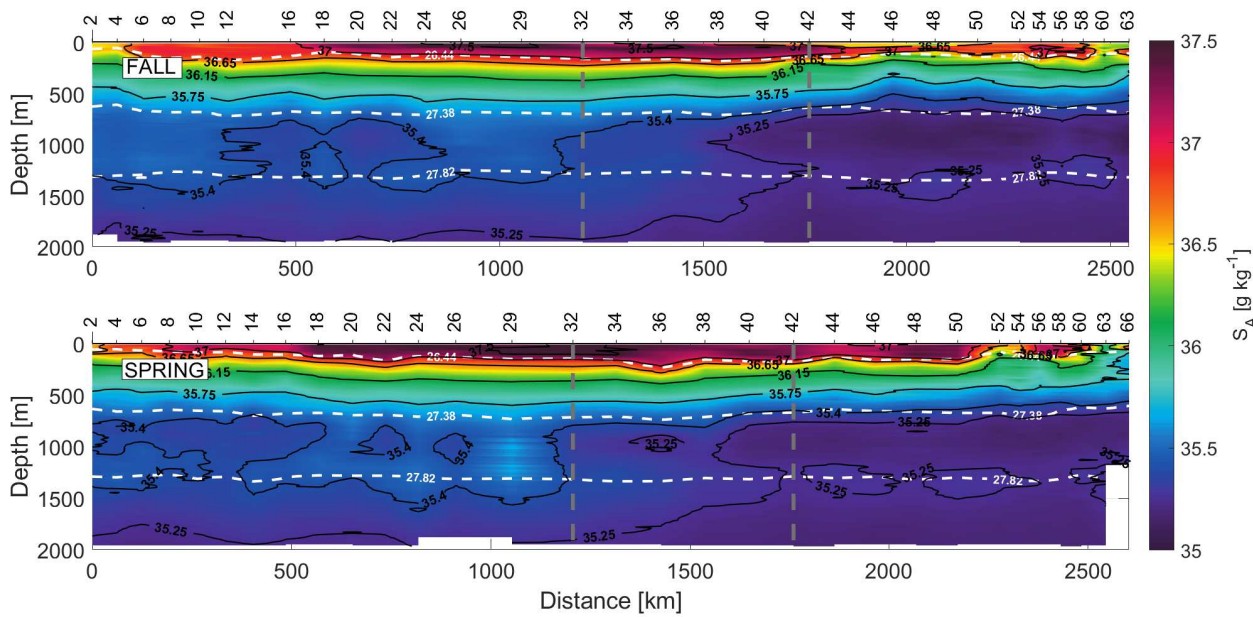

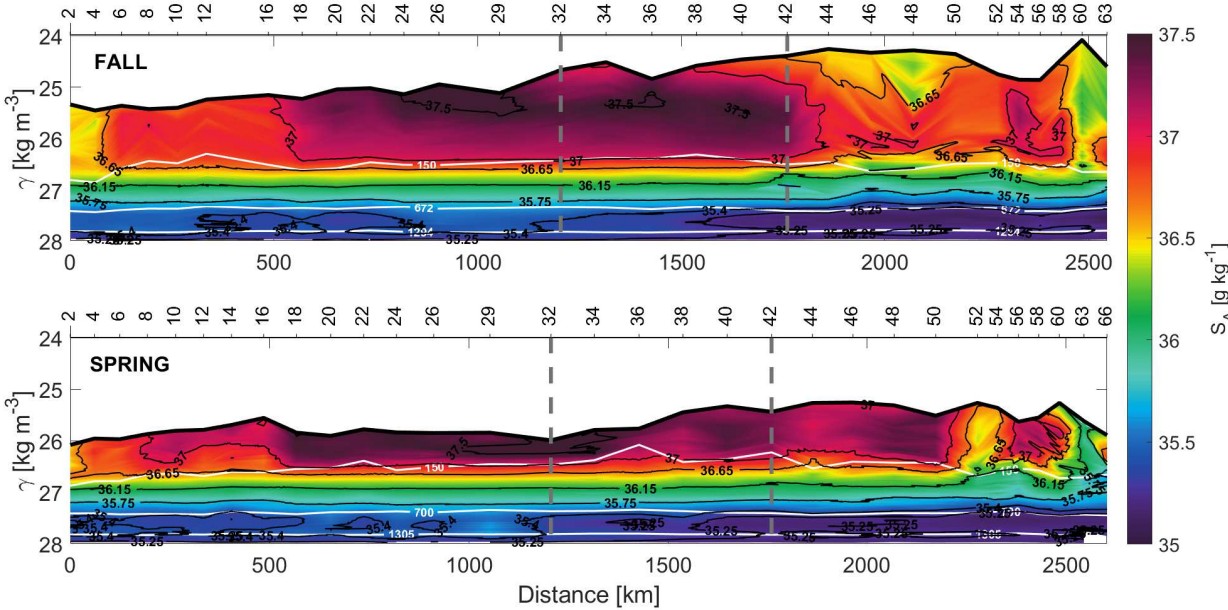

**Figure 4.** Sections of absolute salinity ($S_A$) with respect to depth (top) and $\gamma_n$ (bottom) during fall and spring. In depth section (top), the isoneutrals which delimit the transports at surface, central, intermediate and deep water are represented by white dashed contours. In $\gamma_n$ section (bottom), the depths of 150, 672/700 and 1204/1305 m are also shown.


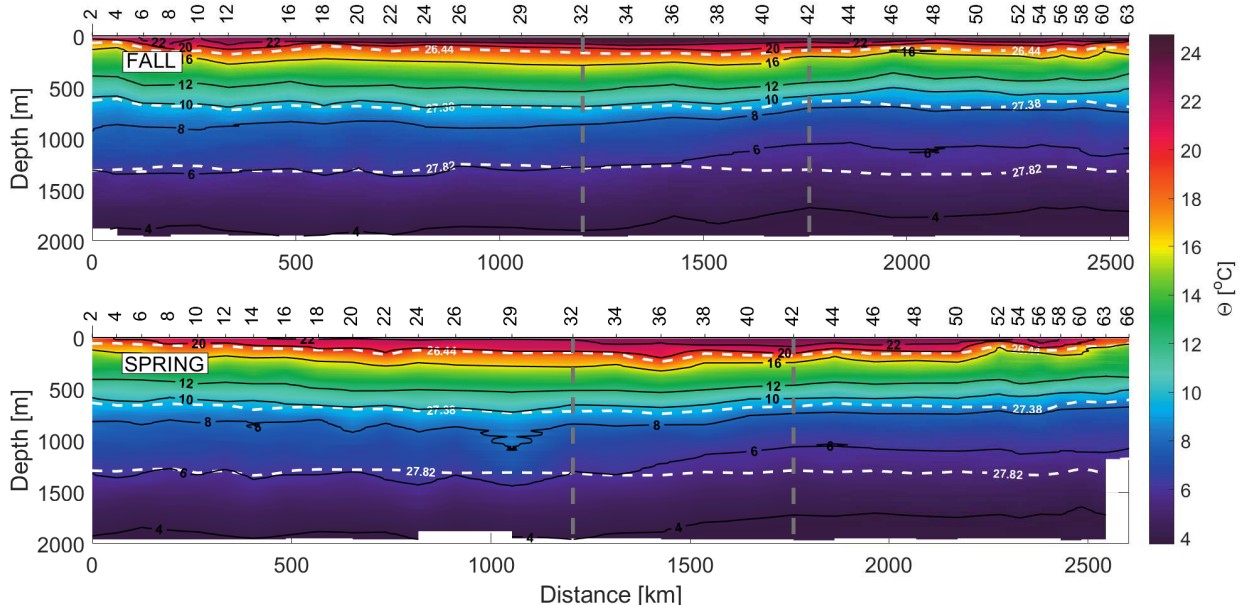

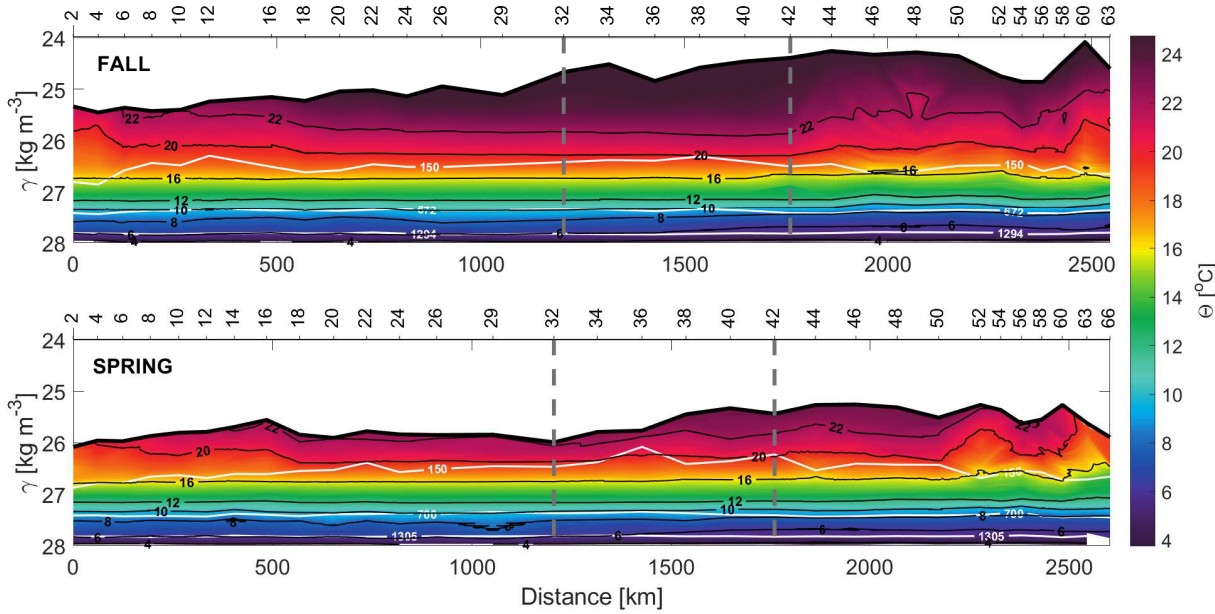

**Figure 5.** Sections of conservative temperature (Θ) with respect to depth (top) and $\gamma_n$ (bottom) during fall and spring. In depth section (top), the isoneutrals which delimit the transports at surface, central, intermediate and deep water in the water column are represented by white dashed contours. In $\gamma_n$ section (bottom), the depths of 150, 672/700 and 1204/1305 m are indicated.



**Figure 6.** Scatter plots for $SiO_2$, $NO_3$ and $PO_4$ nutrients ($\mu mol\,kg^{-1}$ extracted from GLORYS-BIO), and for $DOC$ (observational data in $\mu mol\,L^{-1}$) with respect to $S_A$ and $\gamma_n$ at the north (left), west (middle) and south transects (right) in fall (top) and spring (bottom). The isoneutrals 26.44, 27.38 and 27.82 $kg\,m^{-3}$ that limit the waters layers are indicated with white dashed lines in the colorbar. The measured IN concentrations ($\mu mol\,kg^{-1}$) for $SiO_2$, $NO_X$ and $PO_4$ until 250 m depth are included as black dots.





**Figure 7.** Sections for $SiO_2$ (top), $NO_3$ (middle) and $PO_4$ (bottom) concentrations with respect to $\gamma_n$ during fall (top) and spring (bottom) extracted from GLORYS-BIO. The white isolines as in the $\gamma_n$ sections of Figs. 4 and 5.





**Figure 8.** Sections of DOC concentration with respect to $\gamma_n$ during fall (top) and spring (bottom) cruises with the white isolines as in the $\gamma_n$ sections of Figs. 4 and 5.

**Figure 9.** Reference level velocity at $27.962\,\mathrm{kg\,m^{-3}}$ and its standard deviation estimated by the inverse model during fall (top) and spring (bottom). The direction chosen for the representation is the same as in Fig. 2. The signs of the velocity are according to the geographical criterion, i.e., the velocities are positive/negative toward north/south, in the northern and southern transects and they are positive/negative toward east/west in the western transect.

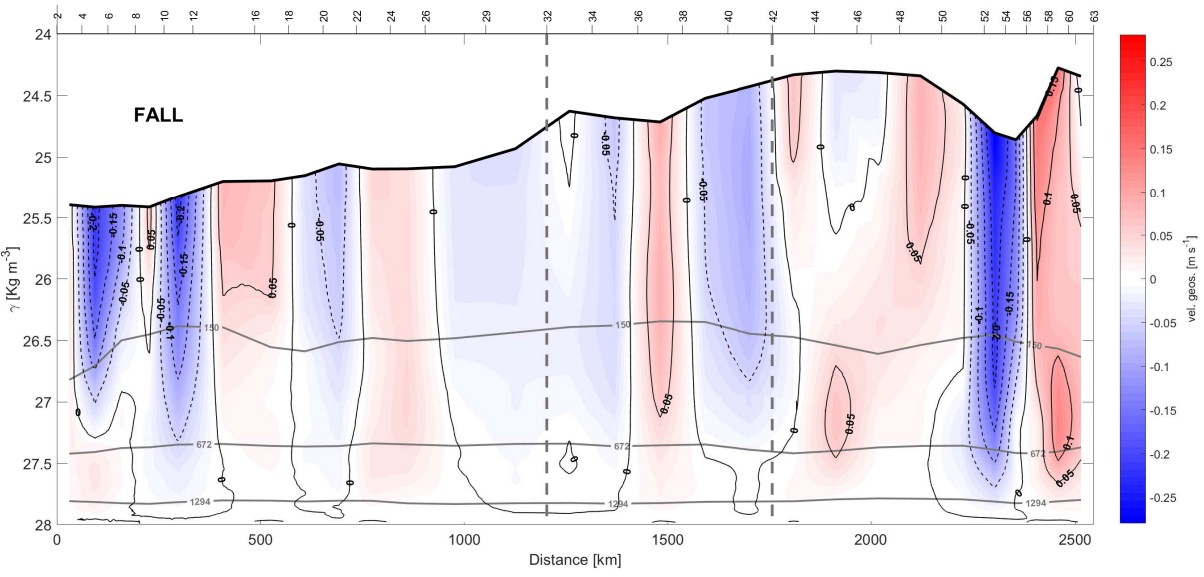

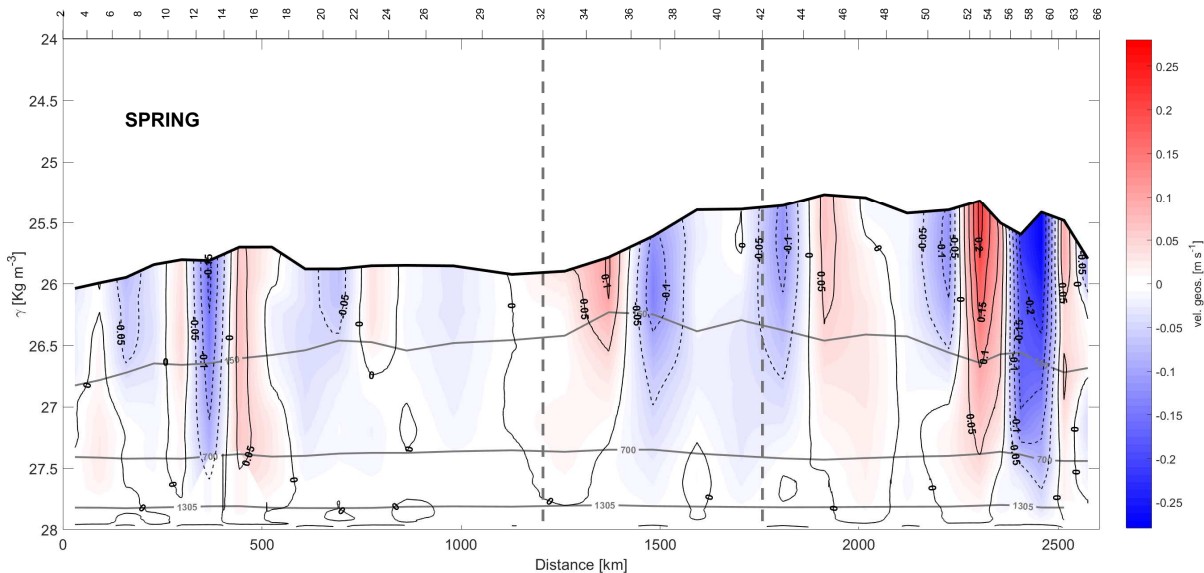

**Figure 10.** Sections of the absolute geostrophic velocity with respect to $\gamma_n$ during fall (top) and spring (bottom). The horizontal axis has the same direction as Figure 2 and the criterion of the velocity signs is as in Figure 9. The depths 150, 672/700 and 1204/1305 m are highlighted by grey isolines as in the $\gamma_n$ sections of Figures 4 and 5.



**Figure 11.** Average derived geostrophic velocity and SLA during fall (top) and spring (bottom) extracted from AVISO+. The red bars represent the mass transports in the shallowest layer as estimated by the inverse model.





**Figure 12.** Accumulated mass transport along the fall (top) and spring (bottom) cruises at surface waters (SW, in red and dashed line), central waters (CW, in red line), intermediate waters (IW, in green line) and deep waters (DW, in blue line). The accumulated mass transport integrated for all the nine layers is also represented. The horizontal axis has the same direction as Fig. 2. Negative/positive values of transports along the three transects indicate inward/outward transports of box delimited by the three transects and the African coast.


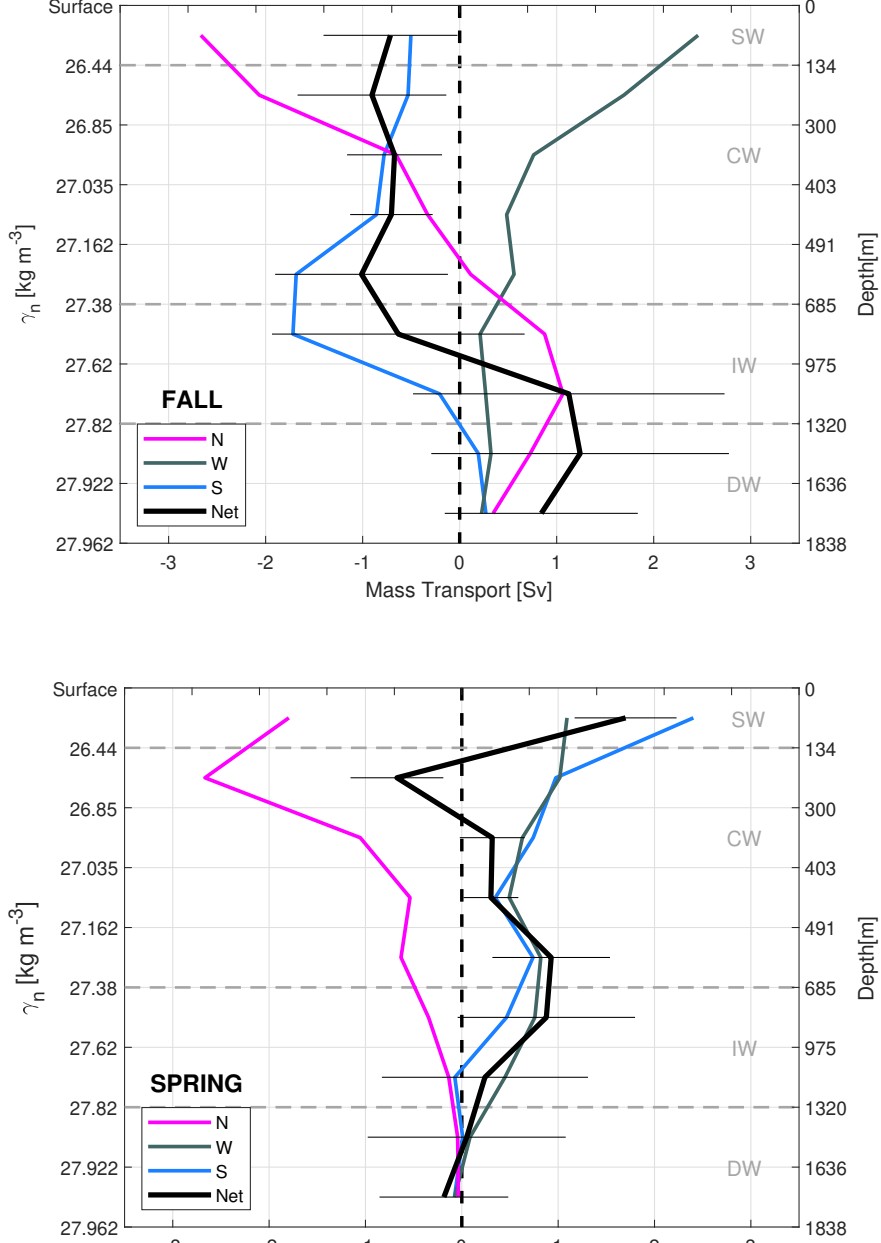

**Figure 13.** Accumulated mass transports per transect at north (N, magenta line), west (W, dark grey line) and south (S, blue line) transects during fall (top) and spring (bottom). SW transport corresponds to transports between surface and the isoneutral 26.44 kg m$^{-3}$, CW transport between the isoneutrals 26.44 and 27.38 kg m$^{-3}$, IW transport between the isoneutrals 27.38 and 27.82 kg m$^{-3}$ and DW transport between the isoneutrals 27.82 and 27.962 kg m$^{-3}$. Negative/positive values indicate inward/outward transports as in Fig. 12. Mass conservation in the whole domain is shown by the black line. The horizontal bars represent the uncertainties estimated by the model.



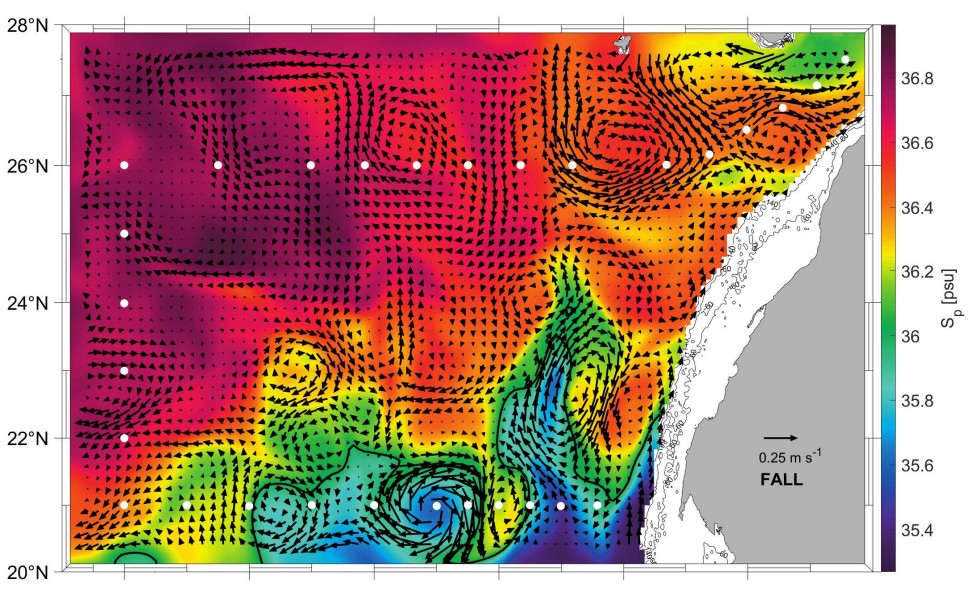

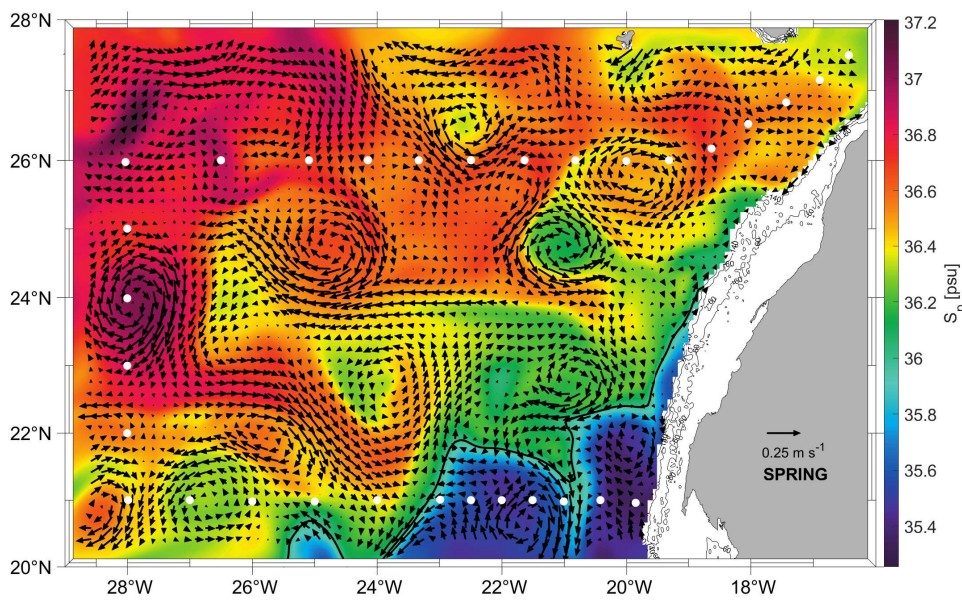

**Figure 14.** Mean salinity and mean geostrophic velocity at 156 m extracted from GLORYS during fall (top) and spring (bottom). The hydrological sampling stations are represented in white dots and the black line indicates the position of the isohaline of 36 at this depth, used to identify the CVFZ.
**Figure 15.** Accumulated $SiO_2$ and $PO_4$ transports ($kmol\,s^{-1}$) at transects north (N, magenta line), west (W, dark grey line) and south (S, blue line) during fall (top) and spring (bottom). SW transport corresponds to transports between surface and the isoneutral $26.44\,kg\,m^{-3}$, CW transport between the isoneutrals $26.44$ and $27.38\,kg\,m^{-3}$, IW transport between the isoneutrals $27.38$ and $27.82\,kg\,m^{-3}$ and DW transport between the isoneutrals $27.82$ and $27.962\,kg\,m^{-3}$. Negative/positive values indicate inward/outward transports as in Fig. 12. The net transport in the whole box is shown by the black line.
**Figure 16.** Accumulated $NO_3$ transports $(\mathrm{kmol\,s^{-1}})$ and accumulated DOC transports $(10^8\,\mathrm{molC\,d^{-1}})$ at transects north (N, magenta line), west (W, dark grey line) and south (S, blue line) during fall (top) and spring (bottom). SW transport corresponds to transports between surface and the isoneutral $26.44\,\mathrm{kg\,m^{-3}}$, CW transport between the isoneutrals 26.44 and $27.38\,\mathrm{kg\,m^{-3}}$, IW transport between the isoneutrals 27.38 and $27.82\,\mathrm{kg\,m^{-3}}$ and DW transport between the isoneutrals 27.82 and $27.962\,\mathrm{kg\,m^{-3}}$. Negative/positive values indicate inward/outward transports as in Fig. 12. The net transport in the whole box is shown by the black line.

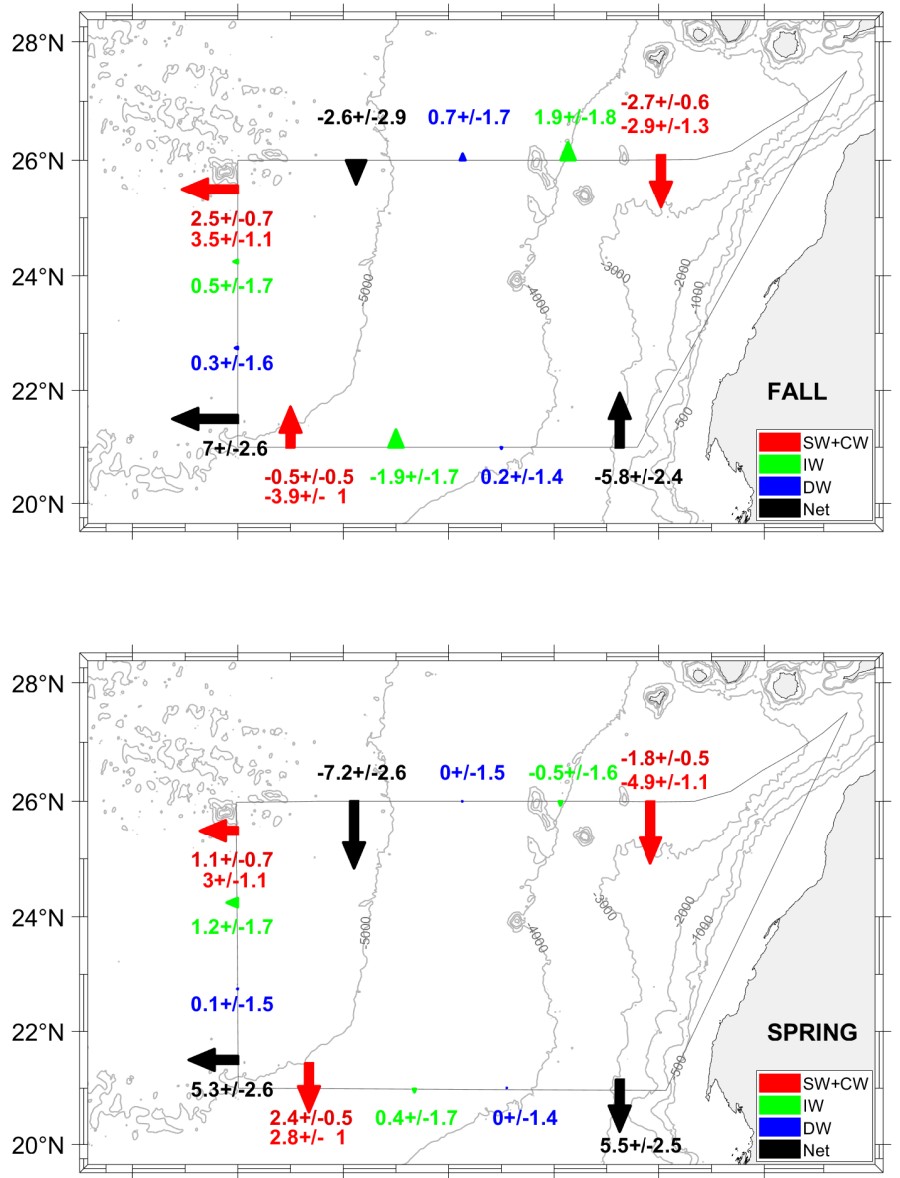

**Figure 17.** Mass transports with their errors (Sv) at surface and central waters (SW+CW, red arrow), intermediate waters (IW, green arrow) and deep waters (DW, blue arrow) across every transect during fall (top) and spring (bottom). Negative/positive values indicate inward/outward transports as in Fig. 12. The arrows in each transect are located in positions which optimize their visibility, representing the integrated transports along each transect. The values of transports at SW (dark red) are given next to the integrated values of transports at CW levels (in red). The red arrows represent the integrated transports for SW plus CW layers.





**Table 1.** Summary of the number and type of measurement in stations per transect and season.

| SEASON [Cruise] | Type of measurement | Number of stations | | | |
|---|---|---|---|---|---|
| | | North | West | South | Total |
| FALL [COCA-I] | CTD | 14 | 6 | 11 | 29 |
| | IN | 8 | 2 | 6 | 14 |
| | DOC | 8 | 2 | 6 | 15 |
| SPRING [COCA-II] | CTD | 15 | 6 | 12 | 31 |
| | IN | 9 | 3 | 8 | 18 |
| | DOC | 10 | 3 | 7 | 18 |





**Table 2.** Summary of water levels (CW, IW, and DW) with their isoneutral limits and their water masses properties for both seasons from the sea surface to 2000 m. The properties extracted from observations are *in situ* temperature (T), potential temperature ($\theta$), conservative temperature ($\Theta$), practical salinity ($S_P$), absolute salinity ($S_A$), and dissolved organic carbon (DOC). IN extracted from GLORYS-BIO are silicates ($SiO_2$), nitrates ($NO_3$) and phosphates ($PO_4$).

| WATER LEVELS | | CW | | | IW | | | DW | |
|---|---|---|---|---|---|---|---|---|---|
| $\gamma_n$ | | MIN. | | MAX. | MIN. | | MAX. | MIN. | MAX. |
| [kg m$^{-3}$] | | 26.44 | | 27.38 | 27.38 | | 27.82 | 27.82 | 27.962 |
| WATER MASSES | | NACW | | SACW | MW | | AAIW | NADW | |
| PROPERTIES | SEASON | MIN. | MAX. | MIN. | MAX. | MIN. | MAX. | MIN. | MAX. | MIN. | MAX. |
| T | FALL | 9.12 | 19.13 | 8.22 | 17.18 | 6.03 | 10.02 | 5.25 | 9.12 | 3.63 | 5.66 |
| [ °C ] | SPRING | 5.90 | 19.76 | 8.35 | 17.14 | 6.01 | 10.04 | 5.16 | 9.41 | 3.63 | 5.57 |
| $\theta$ | FALL | 9.04 | 19.11 | 8.14 | 17.16 | 5.90 | 9.94 | 5.13 | 9.05 | 3.46 | 5.53 |
| [ °C ] | SPRING | 5.77 | 19.74 | 8.27 | 17.13 | 5.88 | 9.96 | 5.06 | 9.34 | 3.47 | 5.45 |
| $\Theta$ | FALL | 9.03 | 19.05 | 8.13 | 17.12 | 5.89 | 9.92 | 5.12 | 9.03 | 3.46 | 5.53 |
| [ °C ] | SPRING | 5.77 | 19.67 | 8.26 | 17.09 | 5.88 | 9.94 | 5.05 | 9.32 | 3.47 | 5.44 |
| $S_P$ | FALL | 35.23 | 36.83 | 35.04 | 36.19 | 35.13 | 35.44 | 34.92 | 35.24 | 34.99 | 35.13 |
| | SPRING | 33.85 | 37.06 | 35.07 | 36.16 | 34.55 | 35.53 | 34.96 | 35.30 | 34.99 | 35.12 |
| $S_A$ | FALL | 35.40 | 37.00 | 35.21 | 36.36 | 35.30 | 35.61 | 35.09 | 35.40 | 35.16 | 35.30 |
| [g kg$^{-1}$] | SPRING | 34.02 | 37.23 | 35.24 | 36.33 | 34.72 | 35.70 | 35.13 | 35.47 | 35.16 | 35.30 |
| $SiO_2$ | FALL | 1.24 | 18.46 | 6.39 | 22.14 | 13.23 | 21.73 | 17.50 | 25.78 | 18.94 | 28.44 |
| [µmol kg$^{-1}$] | SPRING | 1.22 | 21.99 | 6.99 | 23.95 | 13.97 | 21.99 | 17.97 | 28.06 | 19.04 | 28.73 |
| $NO_3$ | FALL | 0.00 | 30.27 | 22.03 | 36.15 | 23.13 | 30.92 | 25.82 | 36.36 | 20.55 | 28.26 |
| [µmol kg$^{-1}$] | SPRING | 0.00 | 30.36 | 25.21 | 36.75 | 23.78 | 31.18 | 25.70 | 36.81 | 21.06 | 27.97 |
| $PO_4$ | FALL | 0.03 | 1.90 | 1.46 | 2.29 | 1.43 | 1.98 | 1.69 | 2.33 | 1.37 | 1.85 |
| [µmol kg$^{-1}$] | SPRING | 0.03 | 1.90 | 1.69 | 2.36 | 1.49 | 1.98 | 1.69 | 2.39 | 1.42 | 1.83 |
| DOC | FALL | 47.85 | 108.65 | 49.05 | 74.13 | 46.25 | 66.09 | 41.83 | 59.30 | 41.82 | 58.72 |
| [µ M] | SPRING | 41.66 | 105.62 | 40.86 | 63.45 | 40.44 | 65.15 | 40.44 | 50.17 | 40.44 | 50.81 |





**Table 3.** *A priori* noise of equations corresponding to SW, CW, and DW levels where the different water masses are transported.

| WATER LEVELS | UNCERTAINTIES ($Sv^2$) |
|:---:|:---:|
| SW and CW | $(1.6 - 4.7)^2$ |
| IW | $(6.3 - 9.3)^2$ |
| DW | $(4.0 - 7.9)^2$ |



**Table 4.** Mass transports with their errors (Sv) for SW, CW, IW, and DW across north, west, and south transects for both seasons. Positive/negative values indicate outward/inward transports. The last row is the integrated transport for all the water column in each transect while the fourth column summarizes the imbalances in mass transport for both seasons.

| WATER LEVELS | SEASON | NORTH | WEST | SOUTH | IMBALANCE |
|---|---|---|---|---|---|
| SW | Fall | $-2.67 \pm 0.60$ | $2.46 \pm 0.66$ | $-0.50 \pm 0.45$ | $-0.71 \pm 1.00$ |
| | Spring | $-1.80 \pm 0.49$ | $1.09 \pm 0.69$ | $2.40 \pm 0.53$ | $1.70 \pm 0.99$ |
| CW | Fall | $-2.94 \pm 1.26$ | $3.50 \pm 1.09$ | $-3.85 \pm 1.03$ | $-3.29 \pm 1.95$ |
| | Spring | $-4.89 \pm 1.14$ | $2.96 \pm 1.06$ | $2.80 \pm 1.02$ | $0.87 \pm 1.86$ |
| IW | Fall | $1.94 \pm 1.85$ | $0.48 \pm 1.71$ | $-1.93 \pm 1.69$ | $0.49 \pm 3.03$ |
| | Spring | $-0.48 \pm 1.65$ | $1.21 \pm 1.68$ | $0.39 \pm 1.73$ | $1.1 \pm 2.92$ |
| DW | Fall | $0.73 \pm 1.71$ | $0.32 \pm 1.56$ | $0.19 \pm 1.37$ | $1.24 \pm 2.69$ |
| | Spring | $-0.04 \pm 1.54$ | $0.09 \pm 1.53$ | $0.00 \pm 1.42$ | $0.05 \pm 2.59$ |
| TOTAL | Fall | $-2.59 \pm 2.88$ | $6.99 \pm 2.64$ | $-5.82 \pm 2.45$ | $-1.43 \pm 4.61$ |
| | Spring | $-7.24 \pm 2.57$ | $5.27 \pm 2.60$ | $5.53 \pm 2.52$ | $3.55 \pm 4.44$ |



**Table 5.** $SiO_2$ transports and their errors $(\mathrm{kmol\,s^{-1}})$ for CW, IW, and DW for north, west and south transects. Positive/negative values indicate outward/inward transports. The last row is the integrated transport in all the water column in each transect and the last column represents the net transport for this variable inside the box.

| WATER LEVELS | SEASON | NORTH | WEST | SOUTH | IMBALANCE |
|---|---|---|---|---|---|
| SW | Fall | $-0.06 \pm 0.01$ | $0.06 \pm 0.02$ | $0.02 \pm 0.02$ | $0.02 \pm 0.02$ |
| | Spring | $-0.06 \pm 0.02$ | $0.04 \pm 0.02$ | $0.06 \pm 0.01$ | $0.04 \pm 0.02$ |
| CW | Fall | $-0.14 \pm 0.06$ | $0.21 \pm 0.06$ | $-0.41 \pm 0.11$ | $-0.34 \pm 0.20$ |
| | Spring | $-0.40 \pm 0.09$ | $0.45 \pm 0.16$ | $0.23 \pm 0.08$ | $0.28 \pm 0.61$ |
| IW | Fall | $0.23 \pm 0.22$ | $-0.13 \pm 0.45$ | $-0.27 \pm 0.24$ | $-0.17 \pm 1.07$ |
| | Spring | $-0.04 \pm 0.15$ | $0.19 \pm 0.27$ | $0.12 \pm 0.55$ | $0.28 \pm 0.72$ |
| DW | Fall | $0.13 \pm 0.31$ | $-0.11 \pm 0.52$ | $-0.14 \pm 1.00$ | $-0.12 \pm 0.25$ |
| | Spring | $-0.01 \pm 0.51$ | $0.06 \pm 1.15$ | $0.08 \pm 13.38$ | $0.13 \pm 6.79$ |
| TOTAL | Fall | $0.16 \pm 0.17$ | $0.03 \pm 0.01$ | $-0.80 \pm 0.34$ | $-0.61 \pm 1.97$ |
| | Spring | $-0.51 \pm 0.18$ | $0.75 \pm 0.37$ | $0.49 \pm 0.22$ | $0.73 \pm 0.91$ |





**Table 6.** $NO_3$ transports and their errors $(\mathrm{kmol\,s^{-1}})$ for CW, IW, and DW for north, west and south transects. Positive/negative values indicate outward/inward transports. The last row is the integrated transport in all the water column in each transect and the last column represents the net transport of this variable inside the box.

| WATER LEVELS | SEASON | NORTH | WEST | SOUTH | IMBALANCE |
|---|---|---|---|---|---|
| SW | Fall | $-0.05 \pm 0.01$ | $0.13 \pm 0.04$ | $0.17 \pm 0.15$ | $0.25 \pm 0.35$ |
|  | Spring | $-0.03 \pm 0.01$ | $0.12 \pm 0.07$ | $0.07 \pm 0.02$ | $0.16 \pm 0.09$ |
| CW | Fall | $-0.36 \pm 0.15$ | $0.47 \pm 0.15$ | $-0.78 \pm 0.21$ | $-0.67 \pm 0.40$ |
|  | Spring | $-0.90 \pm 0.21$ | $0.91 \pm 0.33$ | $0.56 \pm 0.20$ | $0.57 \pm 1.22$ |
| IW | Fall | $0.30 \pm 0.28$ | $-0.16 \pm 0.57$ | $-0.36 \pm 0.32$ | $-0.23 \pm 1.39$ |
|  | Spring | $-0.06 \pm 0.20$ | $0.25 \pm 0.35$ | $0.17 \pm 0.75$ | $0.36 \pm 0.94$ |
| DW | Fall | $0.13 \pm 0.30$ | $-0.10 \pm 0.48$ | $-0.13 \pm 0.91$ | $-0.10 \pm 0.21$ |
|  | Spring | $-0.01 \pm 0.52$ | $0.06 \pm 1.05$ | $0.08 \pm 12.63$ | $0.12 \pm 6.26$ |
| TOTAL | Fall | $0.02 \pm 0.02$ | $0.35 \pm 0.13$ | $-1.11 \pm 0.47$ | $-0.74 \pm 2.40$ |
|  | Spring | $-1.01 \pm 0.36$ | $1.34 \pm 0.66$ | $0.88 \pm 0.40$ | $1.21 \pm 1.51$ |





**Table 7.** $PO_4$ transports and their errors $(\mathrm{kmol\,s^{-1}})$ for CW, IW, and DW for north, west and south transects. Positive/negative values indicate outward/inward transports. The last row is the integrated transport in all the water column in each transect and the last column represents the net transport of this variable inside the box.

| WATER LEVELS | SEASON | NORTH | WEST | SOUTH | IMBALANCE |
|---|---|---|---|---|---|
| SW | Fall | $-0.00 \pm 0.00$ | $0.01 \pm 0.00$ | $0.01 \pm 0.01$ | $0.02 \pm 0.02$ |
| | Spring | $-0.00 \pm 0.00$ | $0.01 \pm 0.01$ | $0.01 \pm 0.00$ | $0.01 \pm 0.01$ |
| CW | Fall | $-0.02 \pm 0.01$ | $0.03 \pm 0.01$ | $-0.05 \pm 0.01$ | $-0.04 \pm 0.02$ |
| | Spring | $-0.06 \pm 0.01$ | $0.06 \pm 0.02$ | $0.04 \pm 0.01$ | $0.04 \pm 0.08$ |
| IW | Fall | $0.02 \pm 0.02$ | $-0.01 \pm 0.04$ | $-0.02 \pm 0.02$ | $-0.01 \pm 0.09$ |
| | Spring | $-0.00 \pm 0.01$ | $0.02 \pm 0.02$ | $0.01 \pm 0.05$ | $0.02 \pm 0.06$ |
| DW | Fall | $0.01 \pm 0.02$ | $-0.01 \pm 0.03$ | $-0.01 \pm 0.06$ | $-0.01 \pm 0.01$ |
| | Spring | $-0.00 \pm 0.04$ | $0.00 \pm 0.07$ | $0.01 \pm 0.85$ | $0.01 \pm 0.42$ |
| TOTAL | Fall | $0.00 \pm 0.00$ | $0.02 \pm 0.01$ | $-0.07 \pm 0.03$ | $-0.05 \pm 0.15$ |
| | Spring | $-0.06 \pm 0.02$ | $0.08 \pm 0.04$ | $0.06 \pm 0.03$ | $0.08 \pm 0.10$ |



**Table 8.** DOC transports and their errors ($10^8 \, \mathrm{mol\,C\,d^{-1}}$) for CW, IW, and DW for north, west and south transects. Positive/negative values indicate outward/inward transports. The last row is the integrated transport in all the water column in each transect and the last column represents the net transport for this variable inside the box. These values are transports of non-refractory DOC which is obtained by subtracting an amount of $40 \, \mathrm{\mu mol\,L^{-1}}$ from the measured DOC.

| WATER LEVELS | SEASON | NORTH | WEST | SOUTH | IMBALANCE |
|---|---|---|---|---|---|
| SW | **Fall** | $-0.37 \pm 0.08$ | $0.04 \pm 0.01$ | $0.42 \pm 0.38$ | $0.10 \pm 0.13$ |
| | **Spring** | $-0.90 \pm 0.24$ | $0.25 \pm 0.16$ | $0.51 \pm 0.11$ | $-0.14 \pm 0.08$ |
| CW | **Fall** | $0.24 \pm 0.10$ | $0.04 \pm 0.01$ | $1.06 \pm 0.28$ | $1.34 \pm 0.80$ |
| | **Spring** | $-1.43 \pm 0.33$ | $0.25 \pm 0.09$ | $0.38 \pm 0.14$ | $-0.80 \pm 1.72$ |
| IW | **Fall** | $0.10 \pm 0.10$ | $-0.03 \pm 0.09$ | $0.04 \pm 0.04$ | $0.12 \pm 0.72$ |
| | **Spring** | $-0.02 \pm 0.08$ | $0.02 \pm 0.02$ | $-0.00 \pm 0.00$ | $-0.01 \pm 0.02$ |
| DW | **Fall** | $0.00 \pm 0.01$ | $-0.00 \pm 0.02$ | $-0.00 \pm 0.00$ | $0.00 \pm 0.00$ |
| | **Spring** | $-0.00 \pm 0.06$ | $0.00 \pm 0.04$ | $0.00 \pm 0.58$ | $0.00 \pm 0.23$ |
| TOTAL | **Fall** | $-0.03 \pm 0.03$ | $0.06 \pm 0.02$ | $1.53 \pm 0.64$ | $1.55 \pm 5.01$ |
| | **Spring** | $-2.35 \pm 0.84$ | $0.52 \pm 0.25$ | $0.89 \pm 0.40$ | $-0.95 \pm 1.19$ |