# Peer review of "Mass, Nutrients and DOC lateral transports off Northwest Africa during fall 2002 and spring 2003"

_Ocean Science, 2019_

## Referee Comment (RC1) · Anonymous Referee #1 · 22 Sep 2019

This work focuses on the seasonal differences in mass and nutrient transports in the upwelling region off northwestern Africa. A inverse model is applied to two sets of closed hydrographic boxes measured in fall 2002 and spring 2003, respectively. The solutions of the inverse model in the two seasons are compared to quantify the differences in horizontal circulation and the resultant mass, nutrient, and DOC transports between fall and spring. The authors have made a lot of efforts to conduct a number of analyses and presented interesting results showing the seasonal differences, which may provide reference for other observational studies and be useful for validation of model outputs.

However, I do not think this manuscript could be published in the current form despite the efforts made by the authors. I would recommend a series of modifications that may to some extent strengthen the conclusion of this work.

**Main comments**

1. One of the main problems I see is that the data used in this study are not sufficient to address the term of "seasonal variability". The authors essentially used two snapshots of hydrographic sections in two different seasons to quantify the "differences" in transports instead of "variability". In my understanding, transport variability can only be discussed when there are continuous timeseries of observations (or model results) or very frequently resampled hydrography, which is not the case for this study. Therefore, I would suggest that the authors may consider (1) to focus on seasonal "difference" instead of "variability"; (2) to use more repeats of the hydrographic sections to increase the samples size (if applicable); and (3) to include and compare with timeseries or/and seasonal cycle of mass, nutrient, and DOC transports from the assimilation models (i.e., GLORYS/GLORYS-BIO) to put the inverse results in a more synthesized context. Please also see the detailed comments below.

2. The authors have performed and listed a large number of analyses including the inverse calculation, and the property transport calculation. However, the "bigger picture" is not very clear and should be improved. The authors mentioned the importance of the EBUS region in association with the southward eastern boundary current of the subtropical/tropical gyre and the CVFZ, which has been studied by many scholars. The authors, however, may emphasize how their study differs from the previous ones, what the new findings are, and why they matter.

3. The discussion appears not well connected with the conclusion. After reading the discussion, I miss how consistent the results in this work are with previous studies. For instance, in lines 414-416, it writes "...This region is featured by a late summer northward progression of AAIW in fall, and by a weak southward flow of MW in spring...".
Whereas, not until lines 464-466 readers would hardly realize the fact that in intermediate layers transport is northward in fall and southward in spring. However, after reading the entire discussion and finally arriving at lines 464-466, readers might have forgotten what was written in the discussion. Therefore, I suggest that the authors merge the discussion and conclusion in one closely related section.

Comments in detail:

1. At the end of the introduction, the authors should point out how this work is different from the previous studies, and why this work is important.

2. Only two repeats of the hydrographic section in fall 2002 and spring 2003 are used in this study. If applicable and convenient, the authors may consider to include more repeats in other years or seasons. This could potentially make this study more representative.

3. Many figures contain subplots. The authors may number the subplots and directly cite the subplots in the text.

4. Line 196. It is not clear what water mass "this last water mass" refers to.

5. The uncertainty of the reference velocity is estimated from the GLORYS velocity, which serves as the a priori error of the unknowns. However, it is not clear whether the reference velocity is taken as 0 everywhere or also estimated from GLORYS. The authors should give information about the reference velocity clearly.

6. Line 232. The deepest common depth is used as the reference depth of each CTD pair. But in case CTD stations are above continental slopes, it should be clarified how the bottom triangle is treated.

7. Line 253. Typo error.

8. The inverse model is constructed without considering vertical (dianeutral) transfer of mass. As the authors stated in the introduction, the EBUS is a constant upwelling re-

OSD
gion due to constant northeasterly winds. What is the influence of the vertical mass flux on the lateral transports? Many of the inverse studies (e.g., Ganachaud 2003; Lumpkin and Speer 2003; Hernandez-Guerra et al., 2005, 2014; Fu et al., 2018) include dianeutral fluxes in different forms in their inverse models, although the dianeutral fluxes are usually not significantly different from 0. It may be convenient for the authors to provide some comments on the sensitivity of inverse solutions to vertical fluxes in the studied region.

9. Line 274. It should be indicated which period is used to calculate the mean SLA.

10. Line 277. It would be better to indicate the exact position of the "remarkable" eddy.

11. Line 327. I assume the "points where the concentrations of DOC are taken" refers to the horizontal position of the stations where DOC are measured? Please indicate that clearly.

12. Throughout the study, the GLORYS/GLORYS-BIO outputs are used. The authors compared the surface layer transport of the inverse solution with AVISO, but they did not show direct comparison of the inverse solution with GLORYS. It is interesting to see to what extent the inverse results agree with the assimilation model, or the other way around. A comparison between the two would serve as a two-way verification. From the assimilation results, timeseries of transports may be calculated and a seasonal cycle may be extracted. These would provide the readers useful information about long-term fluctuation and how representative the inverse estimates are in terms of seasonal, interannual, and long-term variability.

13. If it convenient, the authors may consider to reduce the number of figures. For example, Figures 1 and 14 may be merged.

14. Line 417. Typo error.

---

## Referee Comment (RC2) · Aissa Benazzouz (Referee) · 28 Sep 2019

General CommentsÂă: The authors have tried to analyse the lateral export of nutrients and organic matter in the southern part of the northwest Africa region considered as the highly dynamic upwelling ecosystem.

In this accepted paper, various variables are analyzed in various ways to investigate factors "mainly ocean circulation" controlling the variability in mass, nutrient and DOC. There may be interesting findings, it may help to complete our understanding of the connection between the coastal band and the oligotrophic open ocean in terms of the export of mass and bio-optical proprieties for two distinct periods fall and spring.

[Figure]

Main comments:

1- In my opinion, talking about seasonality seems very exaggerated as long as we have two surveys that have been carrying out during different periods of two different seasons and do not even cover the entire season. In this sense, I prefer that you speak about the comparison of the results obtained during two hydrographics cruises realized at different dates.

2- What happens during summer and winter seasons? Fall and spring are only transitional seasons and the maximum mass, nutrients and DOC lateral transports occur mainly during upwelling seasons summer and winter respectively north and south of Cap blanc where the system is highly dynamic!

3- I believe and I am aware that having hydrographic data covering the whole seasonal cycle is very difficult, but fortunately we have the outputs of the bio-geochemical models and satellites data of the ocean color that can complement your results.

4- Validating the geostrophic flow with SLA in the Figure A9: Superimpose the estimated velocities with SLA can not validate the results, it only gives an idea about the general pattern. I thinks, it's better to make a scatter-plots of the estimated velocity by the inverse model and the derived geostrophic AVISO velocity by transect or all transects can be gathered together for both surveys taken separately.

Minors: The paper can be concise, avoid too much description! The introduction is long, some sentences can be summarized especially in relation to the description of the currents!

Please also note the supplement to this comment:
https://www.ocean-sci-discuss.net/os-2019-91/os-2019-91-RC2-supplement.pdf

---

## Short Comment (SC1) · 21 Oct 2019

We were wrong, the second response for the first reviewer is the answer for the second reviewer. Sorry

---

## Author Comment (AC1) · 21 Oct 2019

We appreciate very much the comments made by the reviewer, that have helped us to produce a clearer version of our manuscript. We have followed them and have introduced several modifications in the paper according to his/her comments. In the next lines we give a detailed repply about how we have handled every comment.

Hereafter, the author's repplies are presented in capital letters.

Main comments:

1. One of the main problems I see is that the data used in this study are not sufficient

to address the term of seasonal variability. The authors essentially used two snapshots of hydrographic sections in two different seasons to quantify the differences in transports instead of variability. In my understanding, transport variability can only be discussed when there are continuous timeseries of observations (or model results) or very frequently resampled hydrography, which is not the case for this study. Therefore, I would suggest that the authors may consider (1) to focus on seasonal difference instead of variability; (2) to use more repeats of the hydrographic sections to increase the samples size (if applicable); and (3) to include and compare with timeseries or/and seasonal cycle of mass, nutrient, and DOC transports from the assimilation models (i.e., GLORYS/GLORYS-BIO) to put the inverse results in a more synthesized context. Please also see the detailed comments below.

WE AGREE WITH THE REFEREE'S COMMENT ABOUT THE SEASONAL NATURE OF THIS MANUSCRIPT. THE TITLE WILL BE CHANGED TO "'MASS, NUTRIENTS AND DOC LATERAL TRANSPORTS OFF NORTHWEST AFRICA DURING FALL 2002 AND SPRING 2003'". THESE CRUISES WERE PERFORMED SOME 16 YEARS AGO, AND THOSE WERE THE ONLY REALIZATIONS AVAILABLE TO BE ANALYSED AS PART OF COCA PROJECT.

THE MAIN STRENGTH IN THIS ANALYSIS IS RELATED TO ITS OBSERVATIONAL NATURE. WE HAVE CHECKED OUT THE HISTORICAL DATABASE AND IT IS REALLY SCARCE IN THIS DOMAIN, WITH LESS THAT 100 STATIONS DURING EACH SEASON AFTER 2001. HENCE, WE CONSIDER THAT RESULTS BASED ON OBSERVATIONS MADE IN THIS PARTICULAR DOMAIN ARE MORE ROBUST THAN RESULTS OBTAINED FROM ASSIMILATION MODELS AS THE HISTORICAL DATABASE THEY ARE BASED ON MIGHT BE UNDERSAMPLED IN THIS DOMAIN.

2. The authors have performed and listed a large number of analyses including the inverse calculation, and the property transport calculation. However, the "bigger picture" is not very clear and should be improved. The authors mentioned the importance of the EBUS region in association with the southward eastern boundary current of the

subtropical/tropical gyre and the CVFZ, which has been studied by many scholars. The authors, however, may emphasize how their study differs from the previous ones, what the new findings are, and why they matter.

THIS STUDY DIFFERS FROM OTHERS WHICH ANALYZE THE SAME AREA BECAUSE SO FAR THE CIRCULATION OF THE EBUS ZONE HAS BEEN STUDIED MAINLY SINCE THE UPWELLING PROCESS ITSELF, BUT STUDIES WHICH RELATE THE MESOSCALAR ACTIVITY AND THE POSITION AND ORIENTATION OF THE CVFZ WITH THE CIRCULATION PATTERNS, OR WITH THE LATERAL ADVECTIVE TRANSPORTS OF BIOGEOCHEMICAL VARIABLES, ARE NOT VERY ABUNDANT IN THE ZONE. WE WILL MODIFY THE CONCLUSIONS IN THE MANUSCRIPT TO HIGHLIGHT THE MAIN OUTCOMES.

3. The discussion appears not well connected with the conclusion. After reading the discussion, I miss how consistent the results in this work are with previous studies. For instance, in lines 414-416, it writes "... This region is featured by a late summer northward progression of AAIW in fall, and by a weak southward flow of MW in spring...". Whereas, not until lines 464-466 readers would hardly realize the fact that in intermediate layers transport is northward in fall and southward in spring. However, after reading the entire discussion and finally arriving at lines 464-466, readers might have forgotten what was written in the discussion. Therefore, I suggest that the authors merge the discussion and conclusion in one closely related section.

WE CONSIDERED THAT TWO SECTIONS MIGHT BE CLEARER IN PRESENTING THE RESULTS. WE TAKE THE COMMENT MADE BY THE REVIEWER AND WILL MODIFY THE DISCUSSION AND CONCLUSIONS SECTIONS TO MAKE THEM MORE UNDERSTANDABLE.

Comments in detail:

1. At the end of the introduction, the authors should point out how this work is different from the previous studies, and why this work is important.

WE AGREE WITH THE REVIEWER AND HAVE MODIFIED THE TEXT AS: "'The ocean dynamics in the region between 20° and 28° N off Northwest Africa during two different seasons is addressed in this manuscript. This domain south of the Canary Islands has historically received less attention as compared to other domains off Northwest Africa, and a proof of that are the few observations available in the historical databases. An inverse box model is applied to hydrographic observations to estimate mass transports. This method provides a velocity field consistent with both mass and properties conservation within a closed volume and with the thermal wind equation (Wunsch, 1996). Several authors have already described the circulation patterns of the NASG by applying an inverse model (Ganachaud and Wunsch, 2002a; Ganachaud, 2003b, a; Hernández-Guerra et al., 2005; Machín et al., 2006; Pérez-Hernández et al., 2013; Hernández-Guerra et al., 2017).

To sum up, the main goal of this manuscript is to present a high quality hydrographic database and to estimate mass, nutrient and organic matter transports during fall and spring seasons south of the Canary Islands in the context of a highly variable environment as the CVFZ. The remaining of this manuscript is 80 organized as follows: the dataset is presented in section 2; the seasonal distribution of the water masses and their properties is displayed in section 3; the technical details of the inverse box model are covered in section 4; the resulting velocity field and the corresponding mass, nutrient and organic matter transports are presented in section 5. Section 6 is devoted to the discussion to end up with some conclusions at section 7."'

2. Only two repeats of the hydrographic section in fall 2002 and spring 2003 are used in this study. If applicable and convenient, the authors may consider to include more repeats in other years or seasons. This could potentially make this study more representative.

WE AGREE WITH THE REVIEWER BUT MORE REPETITIONS WITH THE SAME QUALITY AND DATA DISTRIBUTION ARE NOT AVAILABLE IN THE REGION.

3. Many figures contain subplots. The authors may number the subplots and directly cite the subplots in the text.

WE HAVE MADE AN EFFORT TO BE CLEAR IN THIS ISSUE AND SUBPLOTS CONTAIN THE WORDS "'FALL'" OR "'SPRING'" TO BE PRECISE IN WHAT WE ARE CITING IN THE TEXT. WE HAVE DETECTED THIS TO BE MISSING IN A FEW CASES (Figures 2, 15 and 16) AND HAVE NUMBERED THOSE SUBPLOTS.

4. Line 196. It is not clear what water mass "this last water mass" refers to.

IT REFERS TO SACW. THE TEXT HAS BEEN MODIFIED AND NOW READS AS "'SACW presents maximum...'"

5. The uncertainty of the reference velocity is estimated from the GLORYS velocity, which serves as the a priori error of the unknowns. However, it is not clear whether the reference velocity is taken as 0 everywhere or also estimated from GLORYS. The authors should give information about the reference velocity clearly.

WE HAVE NOW REALIZED AFTER THE REVIEWER'S COMMENT THAT WE DID NOT PRESENT THE REFERENCE LEVEL AS A MOTIONLESS LEVEL. WE HAVE MODIFIED THE TEXT TO CLARIFY THIS ISSUE.

WE HAVE BEEN MODIFIED THE TEXT AS: "'Initially, the reference level is considered as a motionless level where the geostrophic velocity is taken as null before applying the inversion.'"

6. Line 232. The deepest common depth is used as the reference depth of each CTD pair. But in case CTD stations are above continental slopes, it should be clarified how the bottom triangle is treated.

THE SAMPLING IS MADE DOWN TO 2000 M AND, AS IT CAN BE CHECKED OUT IN VERTICAL SECTIONS ON FIGURE 2, THERE IS ONLY ONE STATION OVER THE CONTINENTAL SLOPE DURING THE SPRING CRUISE (NUMBER 66). HENCE, WE ARE NOT GIVING ANY SPECIAL TREATMENT TO THE BOTTOM TRIANGLE

FOUND BETWEEN STATIONS 63 AND 66 AS THE UNCERTAINTY FROM THAT SINGLE BOTTOM TRIANGLE IS LIKELY WITHIN THE UNCERTAINITY OF THE WHOLE ANALYSIS.

7. Line 253. Typo error.

THE TEXT IS CORRECTED AS: "'The velocity variance from the annual mean velocity for each layer is estimated with GLORYS and transformed into (...)'".

8. The inverse model is constructed without considering vertical (dianeutral) transfer of mass. As the authors stated in the introduction, the EBUS is a constant upwelling region due to constant northeasterly winds. What is the influence of the vertical mass flux on the lateral transports? Many of the inverse studies (e.g., Ganachaud 2003; Lumpkin and Speer 2003; Hernandez-Guerra et al., 2005, 2014; Fu et al., 2018) include dianeutral fluxes in different forms in their inverse models, although the dianeutral fluxes are usually not significantly different from 0. It may be convenient for the authors to provide some comments on the sensitivity of inverse solutions to vertical fluxes in the studied region.

WE AGREE WITH THE REVIEWER AND HAVE ADDED THE NEXT TEXT AT THE END OF SECTION 4: "'According to the previous documents north of the Canary Islands, dianeutral velocities are of the order of $10^8$ m s$^{-1}$, while dianeutral diffusion coefficients are of the order of $10^6$ m$^2$ s$^{-1}$ (?). The model results are much less affected by these values than by the reference velocities: a mean dianeutral velocity of $10^8$ m s$^{-1}$ would contribute with only 0.01 Sv, a value much less than the lateral transports obtained from the inverse model. On the other hand, the inverse model provides information only from the box boundaries and cannot be used to infer any detailed spatial distribution of dianeutral fluxes in the coastal transition zone.'".

9. Line 274. It should be indicated which period is used to calculate the mean SLA.

THE MEAN SLA FOR EACH CRUISE IS ESTIMATED WITH SLA FIELDS PRO-

VIDED DURING THE TIME PERIOD THAT EACH CRUISE WAS PERFORMED. THE TEXT IS MODIFIED AS: "'These results are validated by comparison with the surface geostrophic velocity and the sea level anomaly, SLA, derived from altimetry during the time period that each cruise was performed. To do this, the average fields of SLA and geostrophic velocity at the sea surface are calculated during each cruise and shown as a representation of the synoptic situation during both surveys"'.

10. Line 277. It would be better to indicate the exact position of the "remarkable" eddy.

ACTUALLY, AT LINE 277 WE ARE REFERING TO OVERALL MESOSCALE ACTIVITY IN THE SECTIONS. LATER ON, AT LINE 279, WE REFER TO THE PARTICULAR CASE OF A SINGLE EDDY.

11. Line 327. I assume the "points where the concentrations of DOC are taken" refers to the horizontal position of the stations where DOC are measured? Please indicate that clearly.

YES, THE REVIEWER IS RIGHT. THE VELOCITY FIELD IS INTERPOLATED HORIZONTALLY TO THE COORDENATES (LAT,LON) OF THE STATIONS WHERE DOC ARE MEASURED, KEEPING THE DEPTHS, IN WHICH THE VELOCITIES HAVE BEEN CALCULATED. THE TEXT NOW READS AS: "' the velocities are horizontally interpolated to the locations where the concentrations of DOC are taken"'.

12. Throughout the study, the GLORYS/GLORYS-BIO outputs are used. The authors compared the surface layer transport of the inverse solution with AVISO, but they did not show direct comparison of the inverse solution with GLORYS. It is interesting to see to what extent the inverse results agree with the assimilation model, or the other way around. A comparison between the two would serve as a two-way verification. From the assimilation results, timeseries of transports may be calculated and a seasonal cycle may be extracted. These would provide the readers useful information about long-term fluctuation and how representative the inverse estimates are in terms of seasonal, interannual, and long-term variability.

THE FIGURE 14 HAS MODIFIED AND IT IS ATTACHED HERE. THE REVIEWER COULD COMPARE THE INVERSE SOLUTION WITH GLORYS FOR EACH CRUISE.

13. If it convenient, the authors may consider to reduce the number of figures. For example, Figures 1 and 14 may be merged.

WE CONSIDER THAT EVERY FIGURE INCLUDED IN THIS MANUSCRIPT IS REL-EVANT. THE FIRST ONE IS IMPORTANT TO EMPHASIZE THAT THE COLLECTED DATA IS NOT HOMOGENEOUS DURING EACH CRUISE. ON THE OTHER HAND, THE FIGURE 14 IS IMPORTANT TO EMPHASIZE HOW THE POSITION AND ORI-ENTATION OF THE CVFZ AFFECT TO THE TRANSPORTS, BEING ONE OF THE MAIN ISSUES OF THIS STUDY.

14. Line 417. Typo error.

WE HAVE BEEN MODIFIED THE TEXT AS: "'In general, the estimated transport of the three IN shows similar pattern, very marked by the mass transport variability during both seasons"'.
* * *
[Figure]

[Figure]

**Fig. 1.** FALL_GLORYS_VERSUS_MASS_TRANSPORTS

[Figure]

**Fig. 2.** SPRING_GLORYS_VERSUS_MASS_TRANSPORTS

---

## Author Comment (AC2) · 21 Oct 2019

We really appreciate the comments made by the reviewer, that have helped us to produce a clearer version of our manuscript. We have followed them and have introduced several modifications in the paper according to his comments. In the next lines we give a detailed repply about how we have handled every comment.

Hereafter, the author's repplies are presented in capital letters.

Main comments:

1- In my opinion, talking about seasonality seems very exaggerated as long as we

have two surveys that have been carrying out during different periods of two different seasons and do not even cover the entire season. In this sense, I prefer that you speak about the comparison of the results obtained during two hydrographics cruises realized at different dates.

WE AGREE WITH THE REFEREE'S COMMENT. THE TITLE WILL BE CHANGED TO "'MASS, NUTRIENTS AND DOC LATERAL TRANSPORTS OFF NORTHWEST AFRICA DURING FALL 2002 AND SPRING 2003"'.

2- What happens during summer and winter seasons? Fall and spring are only transitional seasons and the maximum mass, nutrients and DOC lateral transports occur mainly during upwelling seasons summer and winter respectively north and south of Cap blanc where the system is highly dynamic!

WE AGREE WITH THE REVIEWER BUT MORE REPETITIONS WITH THE SAME QUALITY AND DATA DISTRIBUTION ARE NOT AVAILABLE IN THE REGION. THE MAIN STRENGTH IN THIS ANALYSIS IS RELATED TO ITS OBSERVATIONAL NATURE AND MAKING AVAILABLE THESE RESULTS WOULD INCREASE THE BACKGROUND KNOWLEDGE IN THE REGION.

3- I believe and I am aware that having hydrographic data covering the whole seasonal cycle is very difficult, but fortunately we have the outputs of the bio-geochemical models and satellites data of the ocean color that can complement your results.

WE INSIST IN OUR IDEA THAT THIS MANUSCRIPT IS RELEVANT BECAUSE OF ITS OBSERVATIONAL NATURE, AND THAT'S THE MAIN MOTIVATION TO PUBLISH THIS DATASET.

4- Validating the geostrophic flow with SLA in the Figure A9: Superimpose the estimated velocities with SLA can not validate the results, it only gives an idea about the general pattern. I thinks, its better to make a scatter-plots of the estimated velocity by the inverse model and the derived geostrophic AVISO velocity by transect or all
transects can be gathered together for both surveys taken separately.

WE PARTIALLY AGREE WITH THE REVIEWER. WE ARE INTERESTED IN AN AV-
ERAGE VALIDATION OF THE RESULTS, AS LONG AS BOTH DATASETS HAVE A
TIME DEPENDENCY SLIGHTLY DIFFERENT FROM EACH OTHER AND A DIRECT
VALIDATION MIGHT BE MISLEADING. IN ANY CASE, WE WILL PRODUCE A NEW
FIGURE TO COMPARE THE VELOCITY FROM THE INVERSION WITH THE VELOC-
ITY FROM THE ALTIMETER DATA.

Minors: The paper can be concise, avoid too much description! The introduction is
long, some sentences can be summarized especially in relation to the description of
the currents!

WE THINK IT IS WORTH PROVIDING DETAILS ABOUT A ZONE THAT HAS HIS-
TORICALLY BEEN UNDERSAMPLED, SOUTH OF THE CANARY ISLANDS. IN ANY
CASE, WE HAVE FULLY PROOFREAD THE MANUSCRIPT TO AVOID ANY REDUN-
DANCY IN THE TEXT AND TO MAKE IT MORE FLUENT AND CLEARER.

---

## Referee Report (RR1)

Main comments:

This manuscript studies the difference in circulation and nutrient transports off the Northwest African Continent between autumn 2002 and spring 2003. The results indicate that due to circulation changes between the two seasons, the studied area works as a nutrient sink (source) in autumn (spring). I think this is an interesting work and the revised manuscript shows improvement, but it still fails to address a key issue that is also raised by both referees in the first round of review. That is the usage of the GLORYS model outputs. It is absolutely true that observational data are very valuable and scarce, but that does not justify the authors' argument that the observational nature of the data is the only important aspect of this work.

First of all, SiO2, NO3, PO4 were so sparsely sampled during both cruises that the authors used the GLORYS outputs to perform the nutrient transport calculation. This highlights the importance of the assimilation model. Secondly, it is a perfect opportunity to show how well the model compares to the observations in terms of circulation and nutrient transports, which provides reference for other studies using the model. This would also potentially be a key strength for this work. Finally, I totally agree with one of the referees in the first round, who wrote: "…*the maximum mass, nutrients and DOC lateral transports occur mainly during upwelling seasons summer and winter respectively north and south of Cap blanc where the system is highly dynamic! ... having hydrographic data covering the whole seasonal cycle is very difficult, but fortunately we have the outputs of the bio-geochemical models…*".

Other comments:

1. The authors should consider to provide a sensitivity experiment on how robust the inverse solutions are with respect to the initial conditions, e.g., reference level, reference velocity, and Ekman transport.

2. Line 18, please indicate what the CINECA program is.

3. Line 85, please indicate what temperature it is, in situ or potential temperature.

4. Line 140, my understanding on this paragraph is that it is about the horizonal difference in salinity among the profiles instead of temporal variability.

5. Line 169-170, here we have a very good example to explain my main point: High values of nutrients are discovered in GLORYS-BIO, and it is attributed to long-lived eddies. Despite the fact that this can be easily verified by showing the GLORYS circulation field and compare with the observations, it is not done.

6. Line 253-255, $10^8$ m s$^{-1}$ is a extremely large number, please check the preciseness.

7. Line 285, what do you mean by vertically shortened? I understand that the water occupied less density range, but in depth space the water is not necessarily thinner in spring than in fall.

8. Line 299, the total mass is not necessarily balanced, because the water column below 2000 m is neglected.

---

## Referee Report (RR2)

The authors have addressed most of my concerns. Particularly, they have made efforts to include a comparison between the observation and the assimilation models by showing seasonal nutrient fluxes at selected points of the three sections. Despite the differences between the assimilation model results and the observation results, the comparison still provides meaningful reference for reader who may be interested in using the model data. Therefore, I think this work deserves to be published after a minor revision.

I have only one minor question:
In the comparison between observed and assimilated nutrient flues, the authors selected one "key point" at each of the northern, western, and southern sections. Could the authors justify their choices of the "key points"? Are these points more representative than other points at each section or than the accumulated nutrient fluxes of each section?

---

## Author Response (AR2)

**Repply to reviewer #1**

We really appreciate the comments made by the reviewer. He/she focuses in the context where these data were gathered and encourage us to include an additional analysis in the manuscript where our results are compared with the outputs from numerical modelling.

Hence, we have included at the end of the discussion some paragraphs where outputs from the GLORYS-BIO are used to estimate nutrient fluxes in the domain. Nutrient fluxes from in situ observations sampled in this manuscript are then compared with those numerical outputs. Both the abstract and the conclusion sections have also been modified to include this new aspect of the manuscript.

**Main comments:**

This manuscript studies the difference in circulation and nutrient transports off the Northwest African Continent between autumn 2002 and spring 2003. The results indicate that due to circulation changes between the two seasons, the studied area works as a nutrient sink (source) in autumn (spring). I think this is an interesting work and the revised manuscript shows improvement, but it still fails to address a key issue that is also raised by both referees in the first round of review. That is the usage of the GLORYS model outputs. It is absolutely true that observational data are very valuable and scarce, but that does not justify the authors' argument that the observational nature of the data is the only important aspect of this work. First of all, $SiO_2$, $NO_3$, $PO_4$ were so sparsely sampled during both cruises that the authors used the GLORYS outputs to perform the nutrient transport calculation. This highlights the importance of the assimilation model. Secondly, it is a perfect opportunity to show how well the model compares to the observations in terms of circulation and nutrient transports, which provides reference for other studies using the model. This would also potentially be a key strength for this work. Finally, I totally agree with one of the referees in the first round, who wrote: "*…the maximum mass, nutrients and DOC lateral transports occur mainly during upwelling seasons summer and winter respectively north and south of Cap blanc where the system is highly dynamic! ... having hydrographic data covering the whole seasonal cycle is very difficult, but fortunately we have the outputs of the bio-geochemical models…*".

**Other comments:**

**1. The authors should consider to provide a sensitivity experiment on how robust the inverse solutions are with respect to the initial conditions, e.g., reference level, reference velocity, and Ekman transport.**

The reviewer is right with respect to the addition of this kind of experiments when working with a priori uncertainties. We have added a reference to a comprehensive analysis performed in a similar domain just north of the one considered in this manuscript.

**2. Line 18, please indicate what the CINECA program is.**

It was been included in the manuscript the full name of the CINECA program.

**3. Line 85, please indicate what temperature it is, in situ or potential temperature.**

It has now been indicated that it is in situ temperature.

**4. Line 140, my understanding on this paragraph is that it is about the horizonal difference in salinity among the profiles instead of temporal variability.**

We disagree with the reviewer. During the fall cruise the upwelling is well developed, a process that doesn't occur during the spring cruise. Hence, the main differences highlighted are related to temporal variability.

**5. Line 169-170, here we have a very good example to explain my main point: High values of nutrients are discovered in GLORYS-BIO, and it is attributed to long-lived eddies. Despite the fact that this can be easily verified by showing the GLORYS circulation field and compare with the observations, it is not done.**

Actually, we are not attributing those high values to long-lived eddies. We humbly suggest that it might be related to long-lived eddies or to variability related to the Cape Verde Frontal Zone.

In any case, GLORYS doesn't provide a velocity field that could explain the variability indicated in the nutrients field.

**6. Line 253-255, $10^8$ ms $^{-1}$ is a extremely large number, please check the preciseness.**

The reviewer is right. We missed a minus sign in the exponent of those numbers. It has been corrected.

**7. Line 285, what do you mean by vertically shortened? I understand that the water occupied less density range, but in depth space the water is not necessarily thinner in spring than in fall.**

The reviewer is right. We have deleted that sentence to avoid any misunderstanding about the depth range affected by the eddies, which is an issue that cannot be concluded from figure 10.

**8. Line 299, the total mass is not necessarily balanced, because the water column below 2000 m is neglected.**

The reviewer is right. Since the box is opened by the bottom, we cannot expect a full imbalance. However, the mass transfer across that bottom layer is expected to be low ($10^{-2}$ Sv), according to the vertical velocities and to the area covered by the cruise. Hence, the mass imbalance must be really close to the values provided in those lines.

---

## Author Response (AR3)

**Dear Editor:**

We are grateful for all the comments made by the reviewer. After giving more importance to numerical modelling outputs and comparing them with the nutrient fluxes obtained from in situ observations, we have included an additional paragraph at the discussion with the aim to explain the criteria used in the selection of the three key points in the comparison of model and observation at each transect. We hope this final change is in accordance with the comment made by the reviewer.

**Repply to reviewer #1**

Regarding the question posed by the reviewer that says:

**In the comparison between observed and assimilated nutrient fluxes, the authors selected one "key point" at each of the northern, western, and southern sections. Could the authors justify their choices of the "key points"? Are these points more representative than other points at each section or than the accumulated nutrient fluxes of each section?**

We have included this paragraph (at line 463) to explain it:

These locations are chosen as the most representative points of each transect. Nutrient profiles at each transect are analyzed for both seasons and the key points are selected where the three nutrients profiles are consistent with their average distribution. In the case of northern and southern transects, the key points are located at the intermediate point between the upwelling area and the oligotrophic open ocean, where the nutrient concentrations remain fairly stable among seasons. At the western transect, the middle position is considered to be the representative point since all nutrients profiles are markedly homogeneous along this transect and, in addition, nutrient concentrations are rather constant from fall to spring (middle column in Fig.6).